# Transsynaptic interactions between IgSF proteins DIP-α and Dpr10 are required for motor neuron targeting specificity

**James Ashley[1], Violet Sorrentino[1], Meike Lobb-Rabe[1,2], Sonal Nagarkar-Jaiswal[3], Liming Tan[4], Shuwa Xu[4], Qi Xiao[4], Kai Zinn[5]\*, Robert A Carrillo[1,2]\***

[1]Department of Molecular Genetics and Cell Biology, University of Chicago, Chicago, United States; [2]Graduate Program in Cell and Molecular Biology, University of Chicago, Chicago, United States; [3]Department of Molecular and Human Genetics, Baylor College of Medicine, Houston, United States; [4]Department of Biological Chemistry, University of California, Los Angeles, Los Angeles, United States; [5]Division of Biology and Biological Engineering, California Institute of Technology, Pasadena, United States

**\*For correspondence:**
zinnk@caltech.edu (KZ);
robertcarrillo@uchicago.edu (RAC)

**Competing interests:** The authors declare that no competing interests exist.

**Abstract** The *Drosophila* larval neuromuscular system provides an ideal context in which to study synaptic partner choice, because it contains a small number of pre- and postsynaptic cells connected in an invariant pattern. The discovery of interactions between two subfamilies of IgSF cell surface proteins, the Dprs and the DIPs, provided new candidates for cellular labels controlling synaptic specificity. Here we show that DIP-α is expressed by two identified motor neurons, while its binding partner Dpr10 is expressed by postsynaptic muscle targets. Removal of either DIP-α or Dpr10 results in loss of specific axonal branches and NMJs formed by one motor neuron, MNISN-1s, while other branches of the MNISN-1s axon develop normally. The temporal and spatial expression pattern of *dpr10* correlates with muscle innervation by MNISN-1s during embryonic development. We propose a model whereby DIP-α and Dpr10 on opposing synaptic partners interact with each other to generate proper motor neuron connectivity.
DOI: https://doi.org/10.7554/eLife.42690.001

## Introduction

The proper wiring of neural circuits is essential for animal behavior, and alterations in connectivity are linked to neurological disease phenotypes in humans (*Rowe, 2010*). Thus, identifying cell-surface molecules involved in neural wiring is critical for understanding biological mechanisms in normal development and in diseased states. Using genetics to uncover these mechanisms has been difficult, partially due to the fact that achieving the necessary precision appears to require partially redundant biochemical interactions.

One of the simplest and most accessible systems in which to study the genetic determination of synaptic connectivity patterns is the *Drosophila* larval neuromuscular system. In each larval abdominal hemisegment, 35 identified motor neurons innervate a set of 30 muscle fibers. Each motor neuron chooses one or more specific muscle fibers as synaptic targets, and the map of connections is almost invariant. *Drosophila* neuromuscular junction (NMJ) synapses are glutamatergic and use orthologs of mammalian AMPA receptors for synaptic transmission. Many scaffolding and regulatory proteins that modulate these receptors are conserved between insects and vertebrates. The sizes and strengths of *Drosophila* NMJs are regulated by retrograde signaling from their postsynaptic muscle targets. In addition to this developmental plasticity, NMJ synapses also exhibit short-term and homeostatic plasticity. These features make the *Drosophila* NMJ a useful genetic model system

for excitatory glutamatergic synapses in the mammalian brain (*Broadie and Bate, 1993*; *Keshishian et al., 1996*; *Menon et al., 2013*).

Although many molecules involved in axon guidance, NMJ morphology, and synaptic activity have been identified through forward and reverse genetic experiments, we still lack an understanding of the mechanisms by which individual larval muscle fibers are recognized as synaptic targets by *Drosophila* motor axons. Gain-of-function (GOF) experiments suggest that individual muscles are labeled by cell-surface proteins (CSPs) that can define them as targets for motor axons. 30 CSPs have been identified that cause motor axons to mistarget when they are ubiquitously expressed in muscles. These proteins contain a variety of extracellular domain (XCD) types, including immunoglobulin superfamily (IgSF) domains and leucine-rich repeat (LRR) sequences. Some of these proteins are normally expressed on subsets of muscles in embryos, suggesting that they could act as molecular signatures during motor axon targeting (*Kurusu et al., 2008*). However, none of the CSPs identified thus far are required for innervation of the muscles that express them, suggesting that they have partially redundant functions (*Chiba et al., 1995*; *Inaki et al., 2007*; *Kurusu et al., 2008*; *Taniguchi et al., 2000*; *Winberg et al., 1998*). In loss-of-function (LOF) mutants lacking CSPs expressed on muscle fibers or the receptors for these proteins on motor axons, innervation occurs normally in most cases. To our knowledge, there are no published LOF mutations in CSP genes that cause high-penetrance failures of innervation of specific muscle fibers.

A network of new candidates for synaptic targeting molecules was recently identified through a global in vitro 'interactome' screen (*Özkan et al., 2013*). In this network, the 'Dpr-ome', a set of 21 proteins with two IgSF domains, the Dprs, interact in a complex pattern with a set of 11 proteins with three IgSF domains, called DIPs (*Cosmanescu et al., 2018*; *Özkan et al., 2013*). We and others have examined the expression patterns of many Dprs and DIPs, and found that each is expressed in a small and unique subset of neurons in the larval ventral nerve cord and pupal brain (*Carrillo et al., 2015*; *Cosmanescu et al., 2018*; *Tan et al., 2015*).

We studied the functions of one Dpr–DIP binding pair, Dpr11–DIP-γ, in both the larval neuromuscular system and the pupal optic lobe. Loss of either *dpr11* or *DIP-γ* produced phenotypes affecting NMJ morphology and retrograde bone morphogenetic protein (BMP) signaling, but did not alter NMJ connectivity patterns. DIP-γ is expressed in most motor neurons, so it is unlikely to be involved in recognition of targets by specific motor neurons. In the optic lobe, however, DIP-γ is selectively expressed in amacrine neurons that are postsynaptic to photoreceptor neurons that express Dpr11, suggesting that Dpr11–DIP-γ interactions may be important in determining synaptic connectivity patterns (*Carrillo et al., 2015*). For several other Dpr–DIP in vitro binding pairs, optic lobe neurons expressing a Dpr are also synaptically connected to neurons expressing the cognate DIP (*Tan et al., 2015*); *Xu et al., 2018*). In the antennal lobe, Dprs and DIPs regulate adhesion and sorting of axons of olfactory receptor neurons (*Barish et al., 2018*).

Based on these findings, we surveyed DIP expression in the larval neuromuscular system, in order to identify DIPs whose expression is restricted to subsets of motor neurons. Remarkably, DIP-α is expressed by only two motor neurons in each hemisegment. There are two types of glutamatergic motor neurons in the larval neuromuscular system: 1b (big boutons) and 1s (small boutons). Larval muscle fibers are divided into four fields: the ventral, ventrolateral, lateral, and dorsal fields. Each 1b motor neuron innervates one or two muscle fibers. The three 1s motor neurons have multiple branches, and each 1s neuron forms branches on most or all of the fibers within a specific muscle field (*Hoang and Chiba, 2001*). DIP-α is expressed in MNISN-1s, which synapses on dorsal muscles, and in MNISNb/d-1s (also referred to as MNSNb/d-1s), which synapses on ventral and ventrolateral muscles. Fate determination and axon guidance of MNISN-1s have been extensively studied in embryos, where it is known as RP2 (*Frasch et al., 1987*; *Landgraf et al., 1997*; *Patel et al., 1989*; *Schmid et al., 1999*; *Sink and Whitington, 1991*).

A subset of muscles innervated by MNISN-1s axon branches are muscles 4, 3, and 2, which are arranged in a ventral→dorsal sequence (*Figure 1C*). In *DIP-α* mutant larvae, the interstitial axon branch onto muscle 4 (m4) is always missing, and the branch onto m3 is usually absent. The branch onto m2, however, is always present. MNISN-1s filopodia are observed in the m4 target area in both wild-type and *DIP-α* mutant embryos, but 1s boutons never form on m4 in mutants. This suggests that nascent axonal projections onto m4 fail to stabilize and convert into NMJs in the absence of DIP-α.

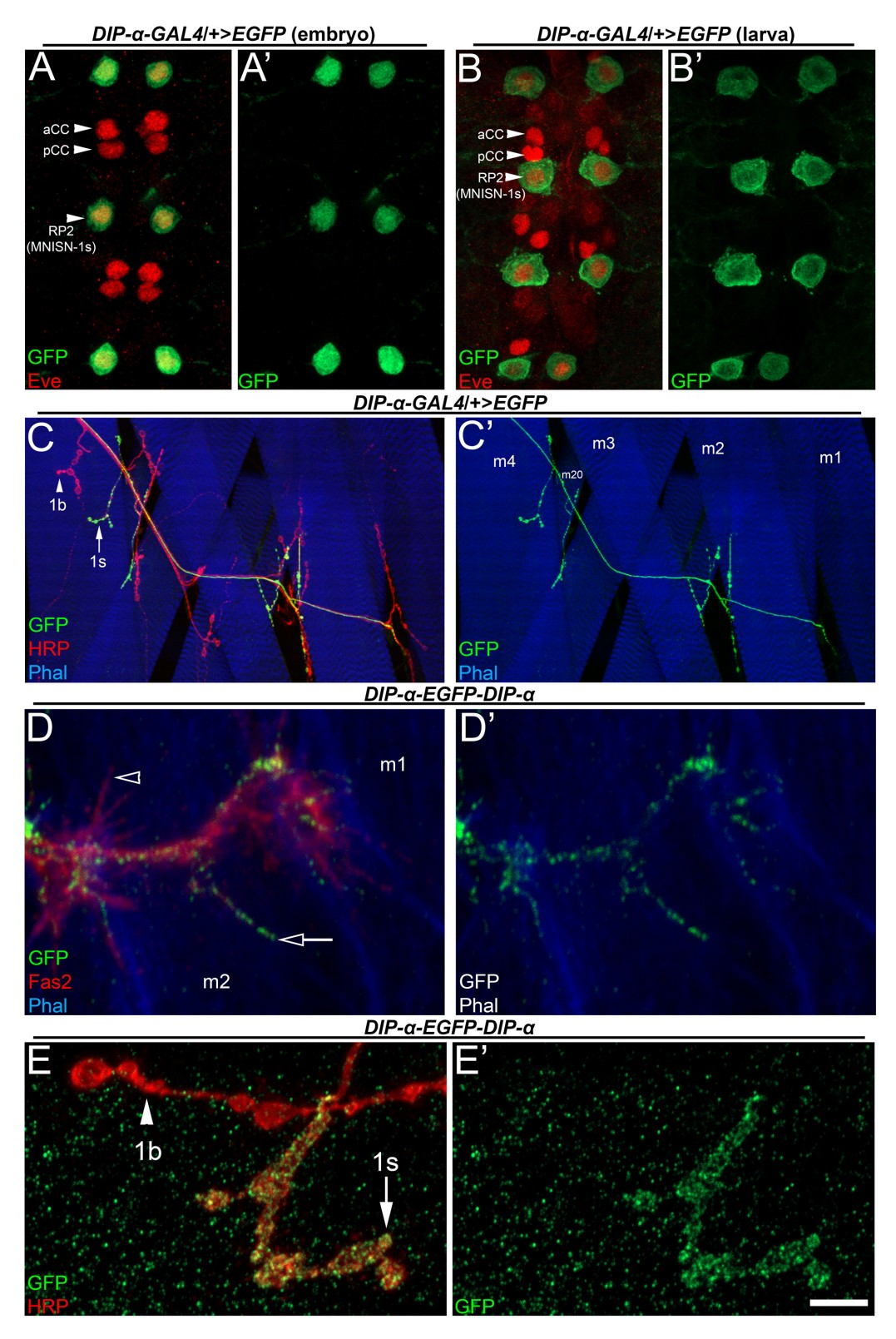

**Figure 1.** DIP-α is expressed in a subset of neurons that includes MNISN-1s/RP2. (A–B) *DIP-α-T2A-GAL4* driving EGFP expression in (A) embryonic and (B) third instar larval ventral nerve cords labeled with anti-GFP (green) and anti-Eve (red). Arrowheads denote segmentally repeating Eve expressing neurons, including RP2/MNISN-1s. (C) Dorsal larval body wall hemisegment labeled with anti-GFP (green), anti-HRP (red) and phalloidin (blue). DIP-α is only expressed in 1s neurons (arrow in C) and not 1b neurons (arrowhead). Muscles are labeled as m1-4 and m20 in C'. (D) *DIP-α-EGFP-DIP-α* protein
*Figure 1 continued on next page*

*Figure 1 continued*

trap embryo labeled with anti-GFP (green) anti-Fas2 (red), and phalloidin (blue), showing that DIP-α is enriched in MNISN-1s filopodia (arrow) but absent in others (arrowhead). (**E**) *DIP-α-EGFP-DIP-α* protein trap reveals DIP-α localization to 1s (arrow) NMJ and not 1b boutons (arrowhead), labeled with anti-GFP (green) and anti-HRP (red). Calibration bar is 16 μm in A, 13 μm in B, 50 μm in C, 3 μm in D and 4 μm in E. See also ***Figure 1—figure supplement 1***.

DOI: https://doi.org/10.7554/eLife.42690.002

The following figure supplement is available for figure 1:

**Figure supplement 1.** DIP-α is expressed in a subset of VNC neurons and DIP-α protein is enriched in the neuropil.

DOI: https://doi.org/10.7554/eLife.42690.003

The 'Dpr-ome' revealed that DIP-α binds to Dpr6 and Dpr10. We examined phenotypes in the larval neuromuscular system caused by loss of these Dprs, and found that in *dpr10* null mutant larvae the MNISN-1s axon branch onto m4 is missing, mimicking the *DIP-α* mutant phenotype. In third instar larvae, *dpr10* is expressed in almost all muscle fibers. However, during motor axon outgrowth in embryos, *dpr10* expression initiates in two muscle fibers in the immediate vicinity of m4, and then comes on in m4 itself around the time at which axon branches appear on this muscle. These results suggest that Dpr10 is a muscle recognition cue whose binding to DIP-α on the motor axon triggers recognition and stabilization of the MNISN-1s filopodia on specific muscles.

The accompanying paper (***Venkatasubramanian et al., 2019***) shows that DIP-α and Dpr10 have expression patterns in adult leg motor neurons and muscles that are qualitatively similar to those seen in the larval neuromuscular system, and that loss of DIP-α or Dpr10 causes failure of DIP-α-expressing leg motor neurons to innervate a subset of their normal muscle targets. Thus, in both of these neuromuscular systems, interactions between DIP-α and Dpr10 control formation of synapses on specific muscle targets.

## Results

### DIP-α is selectively expressed by two identified motor neurons

In a previous study, we showed that several Dprs and DIPs, including *DIP-α,* are expressed in subsets of neurons in the larval ventral nerve cord (VNC). *DIP-α* reporter expression was observed in a subset of neurons (***Figure 1—figure supplement 1B***), including a segmentally repeated pair of motor neurons (***Carrillo et al., 2015***). Here we sought to investigate the identity of these DIP-α-expressing neurons and to determine if *DIP-α* is required for their targeting to specific muscles. We monitored *DIP-α* expression with reporters driven by a gene trap GAL4 in embryos and third instar larvae (***Figure 1A,B***, respectively). The GAL4 line was generated by replacing a splice-trap MiMIC transposable element, MI02031 (***Venken et al., 2011***), in the first *DIP-α* coding intron with a T2A-*GAL4* cassette (***Diao and White, 2012***), using recombination-mediated cassette exchange (RMCE). It has been demonstrated that most MiMIC-derived *GAL4* cassettes faithfully reproduce the expression patterns of the genes into which they are inserted (***Nagarkar-Jaiswal et al., 2015b***). T2A-*GAL4* cassettes in coding introns have the additional feature that GAL4 expression is from a transcript whose translation initiates at the normal ATG of the gene. This means that the expression pattern of GAL4 should correspond to that of the nascent endogenous protein. However, proteins are also subject to post-translational control, so GAL4-driven reporter expression may not necessarily mimic the expression pattern of an accumulated protein as observed by antibody staining (***Diao et al., 2015***; ***Venken et al., 2011***).

The first cells to express the *DIP-α-T2A-GAL4>UAS*-EGFP reporter (*DIP-α-GAL4>EGFP*) include a pair of segmentally repeated neurons in the stage 14 (st14) embryonic VNC (***Figure 3—figure supplement 1A***). *DIP-α* expression persists into late embryonic development, and the segmentally repeated pair of dorsal *DIP-α+* cells also express the transcription factor Even-skipped (Eve) by st16 (***Figure 1A***). The three prominent medially located pairs of Eve+ neurons correspond to the well-characterized aCC, pCC, and RP2 neurons (***Landgraf et al., 1997***; ***Patel et al., 1989***; ***Schmid et al., 1999***). pCC is an interneuron, and aCC is the Ib-type motor neuron that innervates m1. RP2, known as MNISN-1s during larval development (***Figure 1B***), innervates the dorsal muscle field. Based on

the stereotyped positions of the neuronal cell bodies in the VNC that express *DIP-α*, we conclude that *DIP-α* is selectively expressed in MNISN-1s/RP2 (hereafter referred to as MNISN-1s).

In order to prove that the medial *DIP-α+* cell is MNISN-1s, we examined its muscle innervation pattern in third instar larvae. Motor neurons make stereotyped connections with their muscle targets, allowing us to utilize the innervation pattern of the *DIP-α+* Eve+ neuron to identify it. Using the same *DIP-α-GAL4>EGFP* reporter, we observe two axons that exit the VNC in each hemisegment; thus, *DIP-α* expression persists through embryonic and larval development. One of these axons innervates the dorsal muscles (*Figure 1C*), corresponding to the known connectivity map of MNISN-1s (*Hoang and Chiba, 2001*). The other axon innervates ventral and ventrolateral muscles, corresponding to the connectivity map of MNISNb/d-1s (*Figure 1—figure supplement 1C*). The MNISNb/d-1s cell body is located ventrally within the VNC (*Figure 1—figure supplement 1A*) and is not Eve+. When NMJs on all muscles are visualized with anti-HRP, it is observed that only 1s boutons express the *DIP-α-GAL4>EGFP* reporter. Not all 1s boutons are labeled with the reporter, however. There is a third 1s motor neuron that innervates lateral muscles, and this does not express *DIP-α*. In summary, *DIP-α* is expressed by two of the three 1s motor neurons in each abdominal hemisegment, and not by any 1b motor neurons.

DIP-α is a cell surface protein, and analysis of its subcellular localization might provide insights into its function. To address where DIP-α localizes within MNISN-1s, we used a 'protein trap', *DIP-α-EGFP-DIP-α*, constructed by using RMCE to insert GFP into the reading frame of DIP-α, using the same MiMIC as for the T2A-*GAL4* replacement (*Nagarkar-Jaiswal et al., 2015a*). If such protein traps are expressed well, they usually are transported to the subcellular compartments where the endogenous protein is located. During late embryonic development, DIP-α-EGFP-DIP-α localized to growth cones prior to muscle innervation (*Figure 1D*) and remained distributed to all larval MNISN-1s NMJs (*Figure 1—figure supplement 1E*), including m4 (*Figure 1E*). We were unable to determine if DIP-α-EGFP-DIP-α also localizes to MNISN-1s dendrites due to the abundant signal in the VNC neuropil (*Figure 1—figure supplement 1D*).

## DIP-α is required for MNISN-1s targeting specificity

Previously, we and others reported expression patterns that support a role for Dpr-DIP interactions in synaptic partner choice (*Carrillo et al., 2015*; *Tan et al., 2015*). Given the selective expression of *DIP-α* in two 1s motor neurons within the neuromuscular system and the localization of DIP-α to presynaptic terminals, we speculated that *DIP-α* may be involved in targeting of motor axons to muscle fibers. The innervation pattern of the larval musculature is almost invariant, so we can readily evaluate changes in connectivity due to alterations in gene function. MNISN-1s innervates most of the dorsal muscles, including muscles 1, 2, 3, 4, 9, 10, 19, and 20 (*Hoang and Chiba, 2001*).

The *DIP-α-GAL4* reporter that we used for expression studies is also an LOF allele, since the splice-trap into the *GAL4* cassette creates a truncated DIP-α. Conveniently, due to the fact that *DIP-α* is an X-linked gene, we could use *DIP-α-GAL4>EGFP/+* heterozygous females as controls and *DIP-α-GAL4>EGFP/Y* hemizygous males as LOF mutants. In controls, the MNISN-1s branch onto m4, hereafter referred to as m4-1s, is easily identifiable and reproducibly found (*Figure 2A*). Remarkably, loss of *DIP-α* produces a unique phenotype in which the MNISN-1s branch on m4 is always missing (the m4-1s phenotype; *Figure 2B*). In control *DIP-α* heterozygous animals, MNISN-1s innervates m4 with a frequency of 77%, but we never observed m4-1s branches in *DIP-α-GAL4>EGFP* hemizygous males (*Figure 2B,D*). We confirmed the loss of innervation with a targeted null deletion mutation in *DIP-α* (*DIP-α$^{1-178}$*; *Figure 2D*) produced by CRISPR. Thus, DIP-α is necessary for innervation of m4 by MNISN-1s. By contrast, examination of MNISNb/d-1s, the other motor neuron that expresses DIP-α, revealed no obvious highly penetrant phenotype upon loss of *DIP-α*.

## DIP-α is required presynaptically for proper MNISN-1s connectivity

Although *DIP-α* reporters demonstrate that *DIP-α* is expressed in MNISN-1s and localizes to MNISN-1s terminals, we cannot rule out low-level expression in muscles. To define the cells in which *DIP-α* is required for proper MNISN-1s innervation, we performed rescue experiments by expressing a C-terminally Myc-tagged UAS-DIP-α construct in a *DIP-α* mutant background. First, we confirmed that the tagged DIP-α is able to localize to MNISN-1s NMJs on m4 by staining with anti-Myc and anti-DIP-α antibodies (*Figure 2—figure supplement 1A,B*). The lack of DIP-α in the MNISN-1s axon in

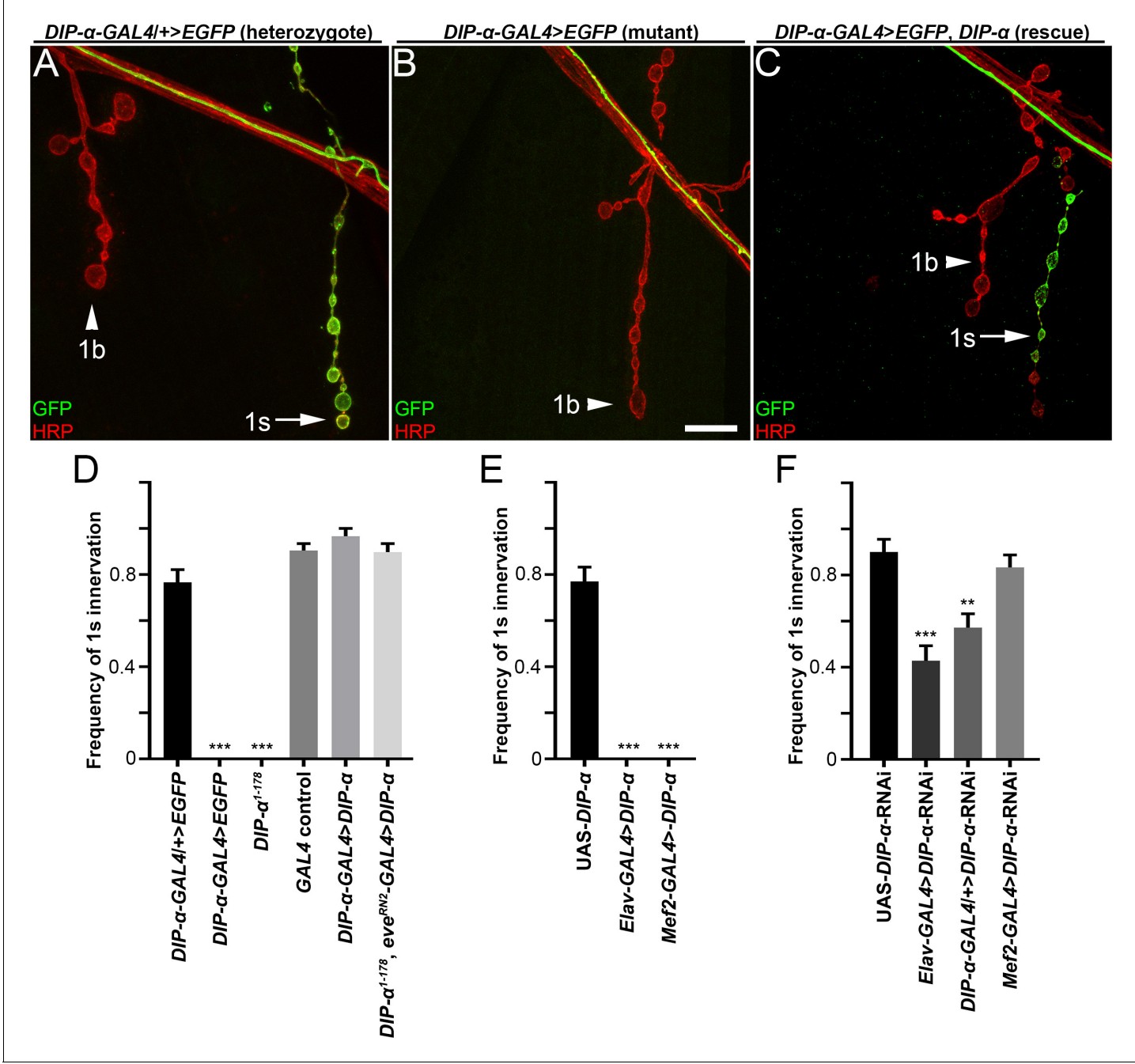

**Figure 2.** DIP-α is required for MNISN-1s branch formation on m4. (A–C) Larval neuromuscular junctions labeled with anti-GFP (green) and anti-HRP (red) from (A) heterozygous *DIP-α-GAL4* animals, (B) homozygous *DIP-α-GAL4* mutants and (C) rescue of homozygous *DIP-α-GAL4* mutants by expressing a UAS-DIP-α construct in DIP-α expressing cells, including MNISN-1s. *DIP-α* is only expressed in 1s neurons (green) (arrows in A,C) and not 1b neurons (arrowhead). (D–F) Frequency of 1s innervation of m4 from (D) mutants and rescue, (E) overexpression of UAS-DIP-α or (F) UAS-*DIP-α*-RNAi. n (animals/hemisegments) = (D) (12/60), (10/30), (18/100), (18/96), (9/56), (14/69), (E) (8/48), (6/30), (9/54), (F) (6/30), (12/71), (10/60), (10/59) (respectively). Calibration bar is 9 μm in A-C. **p<0.001, ***p<0.0001. Error bars represent SEM. See also *Figure 2—figure supplement 1*.

DOI: https://doi.org/10.7554/eLife.42690.004

The following source data and figure supplement are available for figure 2:

**Source data 1.** Data for graphs in *Figure 2*.

DOI: https://doi.org/10.7554/eLife.42690.006

**Figure supplement 1.** DIP-α protein from a transgene localizes normally in rescue animals.

DOI: https://doi.org/10.7554/eLife.42690.005

the intersegmental nerve (*Figure 2—figure supplement 1A,B*) also suggests that DIP-α is actively targeted to the NMJ. We then performed GOF controls to ensure that overexpression of DIP-α in MNISN-1s had no adverse effects on targeting. Overexpression of DIP-α utilizing two drivers that are expressed in subsets of motor neurons (*eve^{RN2}-GAL4*, expressed in MNISN-1s and aCC, plus interneurons; and *DIP-α-GAL4*, expressed in MNISN-1s and MNISNb/d-1s, plus interneurons) produced normal m4-1s NMJs. We expressed these same constructs in a *DIP-α* null background and found complete rescue of the m4-1s phenotype (*Figure 2C,D*), suggesting that *DIP-α* functions in MNISN-1s. We confirmed that DIP-α is required in MNISN-1s using RNAi-mediated knock down of DIP-α. Utilizing either *Elav-GAL4* (pan-neuronal) or *DIP-α-GAL4* to drive RNAi expression, we observed significant reductions in the frequency of m4-1s innervations (*Figure 2F*). Knockdown of DIP-α in muscles using *Mef2-GAL4*-driven RNAi had no effect on m4-1s, indicating that DIP-α is required only in the presynaptic neuron.

Interestingly, driving pan-neuronal expression of DIP-α with *Elav-GAL4* in a wild-type background produced a complete loss of m4-1s, similar to *DIP-α* LOF. The same result was observed when DIP-α was expressed in muscles using *Mef2-GAL4* (*Figure 2E*). Possible models to explain these surprising results are considered in the Discussion.

## DIP-α does not control MNISN-1s axon guidance or defasciculation

Motor neuron axons exit the VNC during embryonic st13-14 and begin to innervate their muscle targets prior to hatching (*Sánchez-Soriano et al., 2007*). During the larval instars, NMJs formed during embryogenesis expand in order to keep pace with the growth of the muscle fibers. We investigated whether *DIP-α* is expressed at the appropriate embryonic stage to function in MNISN-1s muscle targeting. MNISN-1s begins to express *DIP-α-GAL4>EGFP* at st13/14 (*Figure 3—figure supplement 1A*) and this expression continues through embryonic st17 and then persists during larval development (*Figure 1*). The proper wiring of MNISN-1s requires the coordination of axon guidance, fasciculation, defasciculation, and synaptic partner choice. Thus, the striking loss of m4-1s observed upon removal of DIP-α could result from defects in various developmental processes.

Axon guidance requires the integration of attractive and repulsive external cues through cell surface proteins. MNISN-1s innervates the dorsal muscles, the most distal of which is m1. To determine the gross progression of MNISN-1s muscle innervation, we quantified terminal 'swellings' which precede varicosity formation in st15-16 embryos in distal (m1, 2, 9, and 10) and proximal (m3, 4, 19, and 20) muscle fields. MNISN-1s rarely formed terminal swellings in the proximal muscle field at this stage, unlike distal muscles, suggesting that MNISN-1s innervates muscles distal to proximal (*Figure 3—figure supplement 1B*). Thus, if DIP-α is required for axon guidance we would expect to observe defects in MNISN-1s axons in the vicinity of m1 and m2 in mutants. In control st15 embryos, the MNISN-1s growth cone is easily identifiable in the m1/m2 target area (*Figure 3A*). *DIP-α* nulls are indistinguishable from controls, suggesting that axon guidance is likely not affected (*Figure 3B*). Additionally, in third instars, we observe proper MNISN-1s innervation of m2 in *DIP-α* mutants (*Figure 4*).

Another critical step in establishing neuromuscular wiring is the defasciculation of motor neuron axons from the main nerve. This process requires the appropriate balance of adhesion between axons and repulsive signals that facilitate separation of axons from the nerve bundle. Defasciculation is typically initiated by filopodia emanating from the growth cone (when an entire axon leaves the bundle) or from the axon shaft (when an interstitial branch is being formed). We investigated whether *DIP-α* was required for MNISN-1s filopodia to leave the nerve bundle. In control st17 embryos, there are obvious motor neuron filopodia, including MNISN-1s filopodia (identified by their selective expression of *DIP-α-GAL4>EGFP*) that defasciculate from the ISN throughout the dorsal body wall (*Figure 3C*). We observed similar filopodial projections in age-matched *DIP-α* null embryos (*Figure 3D* caret), ruling out a function for DIP-α in general defasciculation (*Figure 3C,D* asterisk).

We then looked more closely at MNISN-1s axon defasciculation and bouton formation at m4, since this would address whether the m4-1s branch initially forms and then retracts or never forms at all in *DIP-α* mutants. During axon pathfinding, filopodia extend and contact both target and non-target muscles. The developmental time course for MNISN-1s innervation of m4 is not known, so we examined MNISN-1s axons at various embryonic stages and in first instar larvae. In control embryos, we observe MNISN-1s axon defasciculation and filopodia at m4 beginning in st17 (*Figure 3C*).

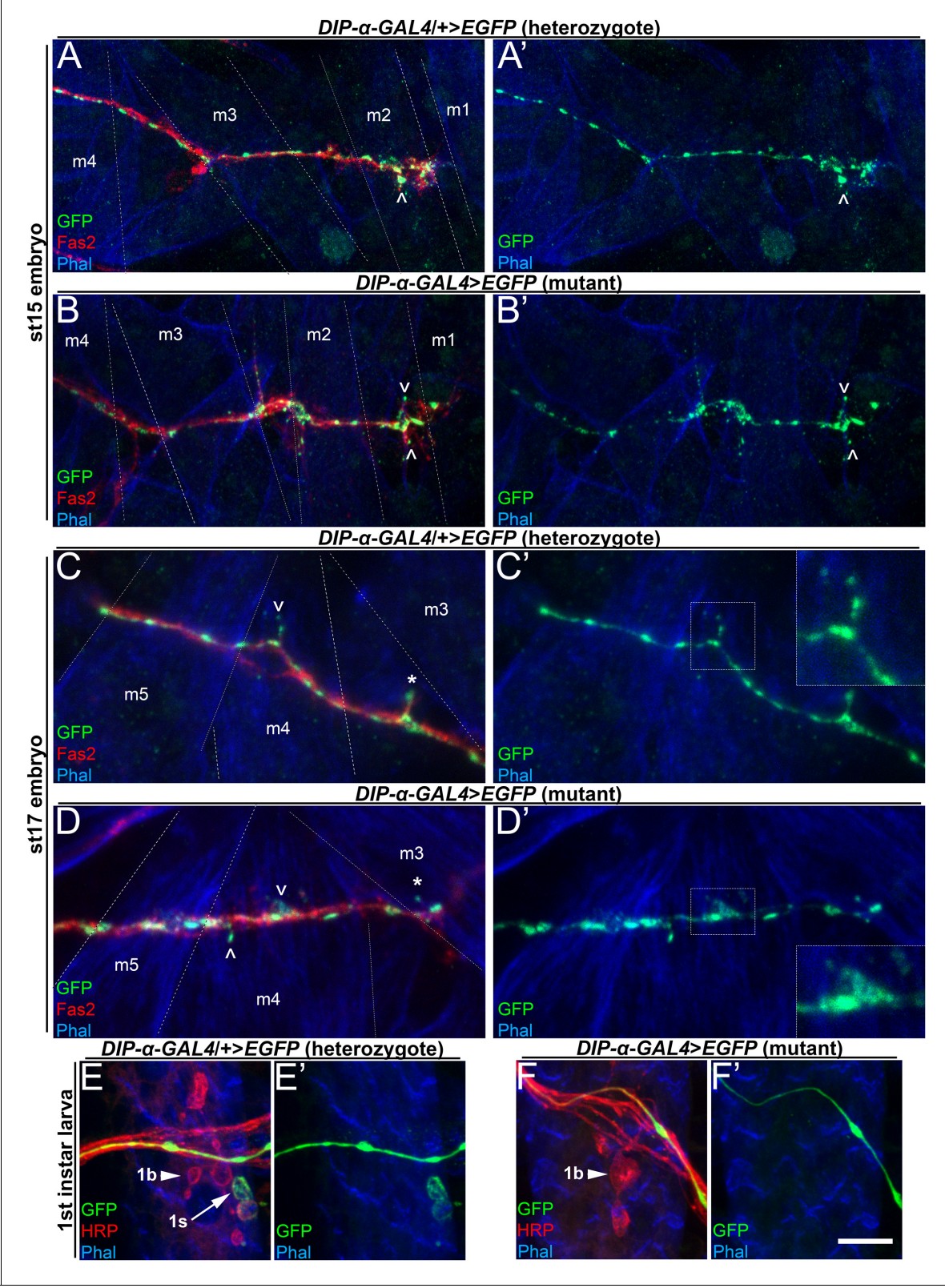

**Figure 3.** MNISN-1s axons in *DIP-α* mutants produce protrusions on m4 but are unable to form stable 1s branches. (**A–D**) Embryonic NMJs labeled with anti-GFP (green), anti-HRP (red) and phalloidin (blue) from (**A,C**) heterozygous and (**B,D**) hemizygous *DIP-α-GAL4* mutants. (**A–B**) are st15 embryos; (**C–D**) are st17 embryos. MNISN-1s protrusions over m4 form late in embryonic development (**C**) while protrusions over m2 form earlier (**A**). Note that MNISN-1s axon protrusions form properly over dorsal muscles, including m2, at st15 in (**A**) controls and (**B**) *DIP-α* mutants and over m4 at st17 in

*Figure 3 continued*

controls (C) and *DIP-α* mutants (D). Also, ˆindicates all filopodia in (A) and (B), and m4 filopodia in (C) and (D); * indicates st17 non-m4 filopodia. (E–F) First instar larva labeled with anti-GFP (green), anti-HRP (red) and phalloidin (blue) from *DIP-α-GAL4* heterozygous animals and (F) *DIP-α-GAL4* homozygous mutants. Note 1b boutons present in both first instar larvae (arrowhead in E,F), but only heterozygous animals have 1s boutons (arrow in E). Calibration bar is 8 μm in A,B, 4 μm in C,D and 6 μm in E,F. See also *Figure 3—figure supplement 1*.

DOI: https://doi.org/10.7554/eLife.42690.007

The following source data and figure supplement are available for figure 3:

**Source data 1.** Data for graphs in *Figure 3—figure supplement 1*.
DOI: https://doi.org/10.7554/eLife.42690.009
**Figure supplement 1.** DIP-α is expressed in MNISN-1s motor neurons in st14 embryos, and 1s arbors form over distal muscles before proximal.
DOI: https://doi.org/10.7554/eLife.42690.008

MNISN-1s filopodia at more dorsal muscles are evident at earlier stages (*Figure 3A*). In *DIP-α* LOF st17 embryos, we still observe MNISN-1s axon protrusions at m4 (*Figure 3D*), suggesting that the m4-1s branch initially extends but is not properly stabilized due to loss of *DIP-α*. In first instar larvae, 1s boutons are found on m4 in wild-type (*Figure 3E*), but we never observed a 1s bouton on m4 in *DIP-α* mutants (*Figure 3F*; 64 hemisegments examined). These data suggest that MNISN-1s filopodia still contact m4 in the absence of DIP-α, but fail to establish a stable interstitial branch.

## Loss of DIP-α selectively reduces and expands MNISN-1s connectivity

The localization of DIP-α to all MNISN-1s boutons prompted us to perform a more exhaustive analysis of MNISN-1s targeting to determine if there were any subtle changes in connectivity. Previous studies reported that MNISN-1s innervates eight dorsal muscles (1, 2, 3, 4, 9, 10, 19, and 20) (*Hoang and Chiba, 2001*). A detailed analysis of innervation shows that MNISN-1s targets m2, 3, 4, 9, 10, 19, and 20 almost invariantly (*Figure 4A,C*). However, m1, the most dorsal muscle, is usually not innervated in control animals.

We repeated this analysis in a *DIP-α* null background and observed interesting alterations in connectivity. First, m1, which rarely has 1s innervation, consistently had 1s branches (*Figure 4B,C*). Second, two muscles, m11 and m18, which are never innervated by MNISN-1s in controls, now sometimes receive ectopic 1s branches upon removal of *DIP-α*. Third, muscles adjacent to m4 also lose 1s innervation in mutants (*Figure 4B,C*). Specifically, m3, 19, and 20 show major reductions in 1s branch frequency. It is interesting to note that the more dorsal muscles in the dorsal muscle field gain 1s branches (m1, 2, 11, 18), while more ventral muscles (m3, 4, 19, 20) lose innervation. m2, for example, has an approximately 2-fold increase in the number of MNISN-1s branches (*Figure 4—figure supplement 1A*).

## Expression of DIP-α binding partners, Dpr6 and Dpr10, in the neuromuscular system

DIP-α binds to only two of the 21 Dprs, Dpr6 and Dpr10 (*Cosmanescu et al., 2018*; *Özkan et al., 2013*). We sought to determine where *dpr6* and *dpr10* are expressed, in order to ascertain if either could potentially function with DIP-α to control MNISN-1s targeting. *dpr6-T2A-GAL4>EGFP* shows *dpr6* expression in subsets of cells in the embryonic VNC, including MNISN-1s and aCC (*Figure 5A*), and analysis of the larval bodywall reveals that *dpr6* is expressed in type 1b and 1s motor neurons (*Figure 5B*). Thus, *dpr6* and *DIP-α* are co-expressed in MNISN-1s.

We next investigated *dpr10* expression using a *dpr10-T2A-GAL4* reporter. *dpr10* is found in subsets of cells in the embryonic VNC (*Figure 5—figure supplement 1A*) but, unlike *dpr6* and *DIP-α*, it is not expressed in embryonic MNISN-1s (*Figure 5C*). *dpr10* is prominently expressed in third instar larval muscles, including m4 (*Figure 5D*). As the GFP signal is very high within muscles, we were unable to determine if *dpr10* was expressed in motor neurons with the current reporter. Some signal is detected in the nerves (*Figure 5D* caret), suggesting that *dpr10* is expressed in neurons that have axons in the periphery. To define Dpr10 localization, we expressed a UAS-Dpr10-V5 construct in muscles. Remarkably, V5 staining was selectively found at NMJs, surrounding both 1b and 1s boutons, indicating that the protein localizes to the postsynaptic side of NMJs (*Figure 5E*; see *Figure 5—figure supplement 1B* for anti-V5 staining of control NMJs).

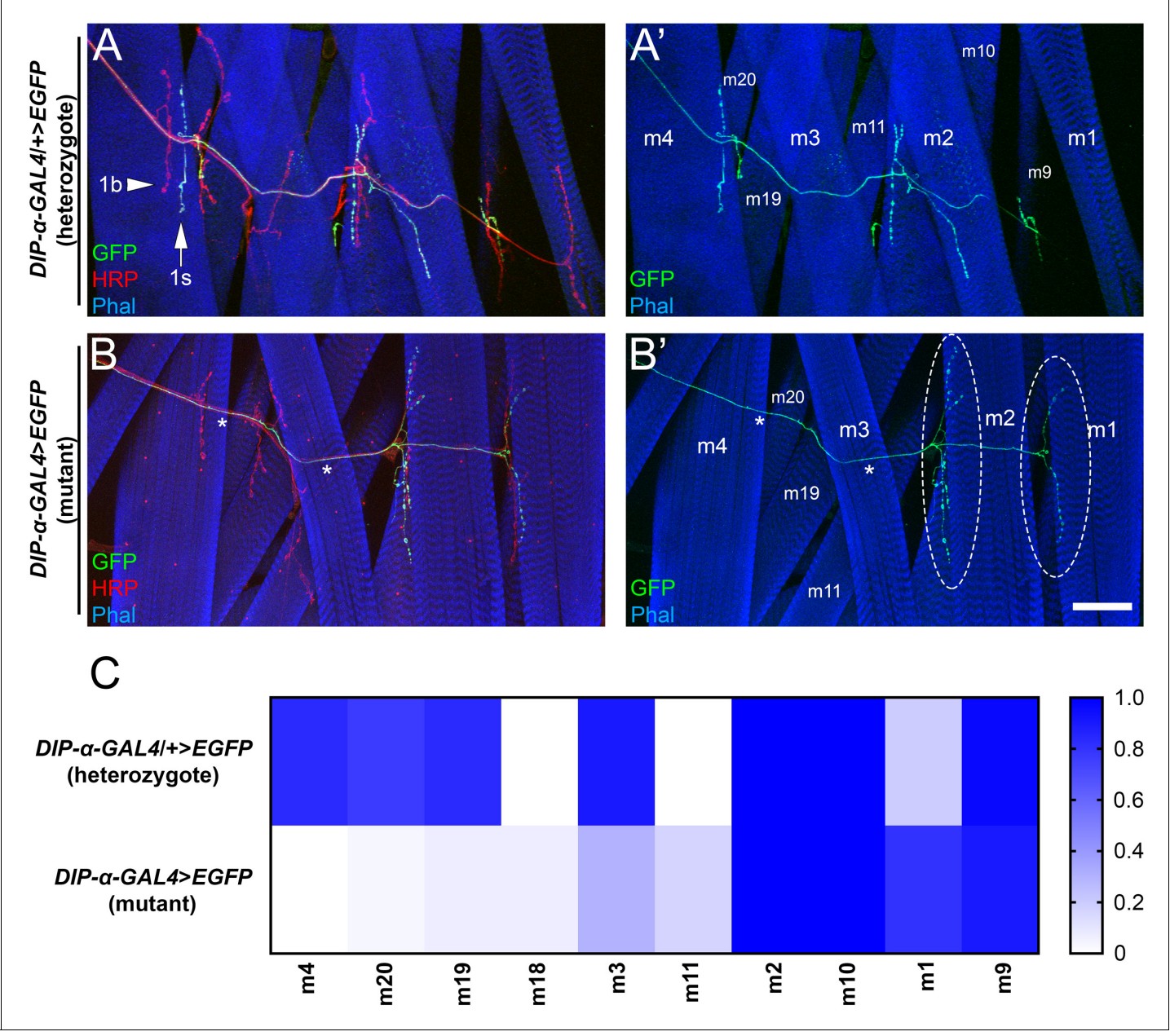

**Figure 4.** Loss of *DIP-α* causes selective loss of MNISN-1s branches on proximal muscles and ectopic innervation of distal muscles. (A,B) Dorsal body wall hemisegments labeled with anti-GFP (green), anti-HRP (red) and phalloidin (blue) from (**A**) heterozygous *DIP-α-GAL4* animals, and (**B**) homozygous *DIP-α-GAL4* mutants. Arrow denotes 1s boutons in (**A**), arrowhead denotes 1b boutons, and * denote muscles in mutant that have lost 1s innervation. Overgrown 1s arbors on m2 and m1 are circled in mutants. (**C**) Quantification of the frequency of innervation of MNISN-1s neurons on the respective muscles from the above genotypes. n (animals/hemisegments) = (16/30), (12/30) (respectively). Calibration bar is 60 μm. See also *Figure 4—figure supplement 1*.

DOI: https://doi.org/10.7554/eLife.42690.010

The following source data and figure supplement are available for figure 4:

**Source data 1.** Data for graphs in *Figure 4* and *Figure 4—figure supplement 1*.
DOI: https://doi.org/10.7554/eLife.42690.012
**Figure supplement 1.** Quantification of MNISN-1s arbor branches on m2.
DOI: https://doi.org/10.7554/eLife.42690.011

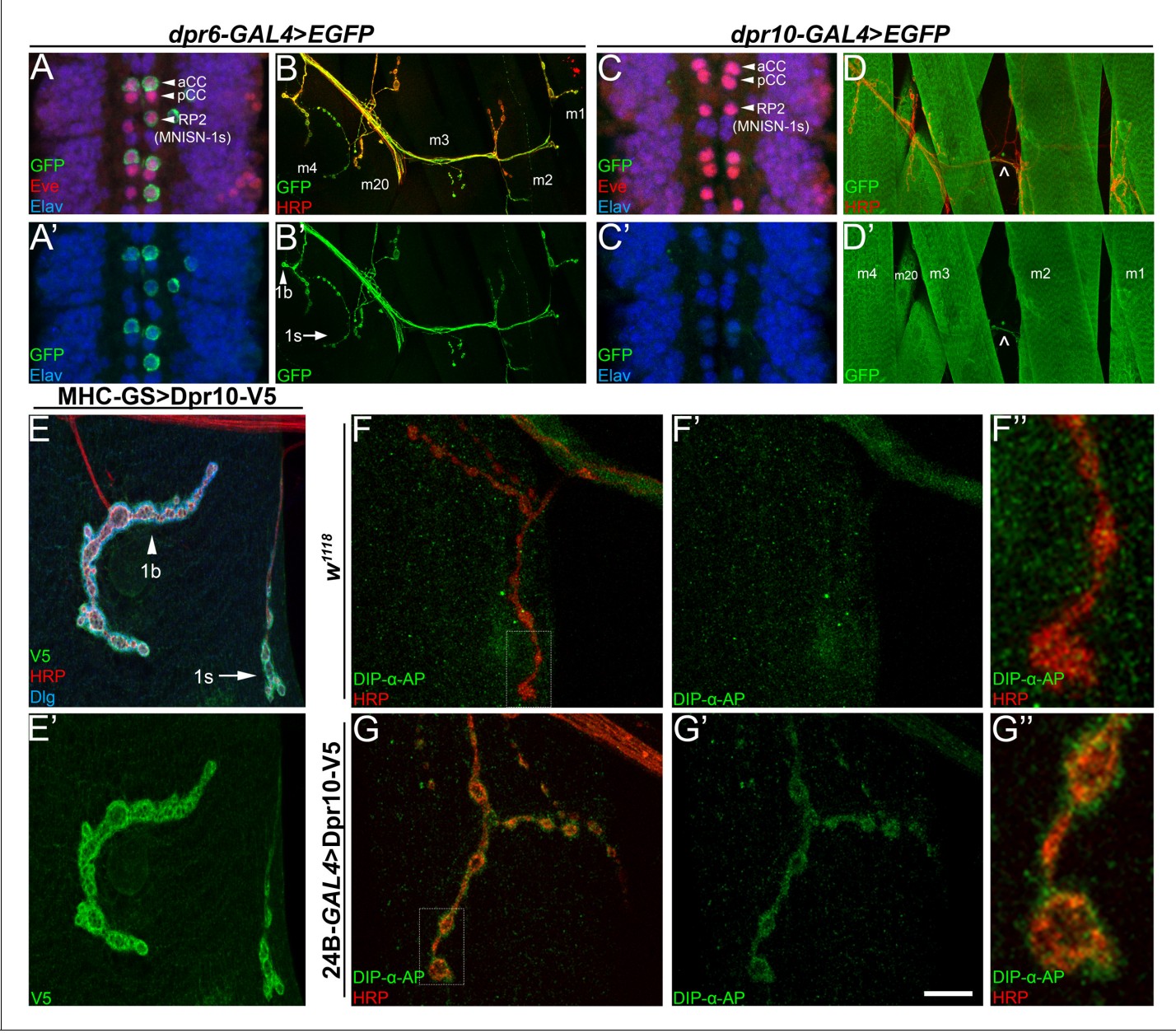

**Figure 5.** Dpr10 is expressed in muscles and can localize to the SSR and bind to DIP-α. (A,C) Embryonic ventral nerve cord from (A) *dpr6-GAL4* and (C) *dpr10-GAL4* driving EGFP expression, labeled with anti-GFP (green), anti-Eve (red) and anti-Elav (blue). *dpr6* is expressed in aCC and RP2 (MNISN-1s) neurons, while *dpr10* is not expressed in these Eve+ cells (see also *Figure 5—figure supplement 1A*). (B,D) Larval dorsal body wall hemisegments from (B) *dpr6-GAL4* or (D) *dpr10-GAL4* driving EGFP expression, labeled with anti-GFP (green), and anti-HRP (red). Muscles are labeled m4-m1, m20. Dpr10 is also expressed in unknown neurons (ˆ in 5D denotes axon). (E) Larval m4 NMJ from animals expressing Dpr10-V5 using *MHC-GeneSwitch-GAL4* in the absence of RU486 (low-level muscle expression only), labeled with anti-V5 (green), anti-HRP (red) and anti-Dlg (blue). 1b and 1s neurons are marked with arrowhead and arrow, respectively. (F,G) Larval m4 NMJ from either (F) control *w^1118* or (G) *24B-GAL4>UAS-Dpr10-V5* (high level pan-muscle expression). Live tissue was incubated with DIP-α-AP5 protein (see Materials and methods) and then labeled with anti-AP (green) and anti-HRP (red). Calibration bar is 15 μm in A,C, 60 μm in B,D, 12 μm in E, and 18 μm in F,G. See also *Figure 5—figure supplement 1*.

DOI: https://doi.org/10.7554/eLife.42690.013

The following figure supplement is available for figure 5:

**Figure supplement 1.** Dpr10 is expressed in a subset of neurons in the embryonic ventral nerve cord, and anti-V5 staining in a control sample.

DOI: https://doi.org/10.7554/eLife.42690.014

## Dpr10 is able to bind DIP-α in tissue

Based on DIP-α localization to MNISN-1s terminals, we hypothesize that DIP-α interacts in trans with another cell surface protein on m4 to confer specificity. An attractive candidate for this role is Dpr10, which is expressed in muscles and interacts biochemically with DIP-α. We sought to determine if DIP-α–Dpr10 interactions could occur in the larval tissue. We used a recombinant DIP-α fused with pentameric alkaline phosphatase (DIP-α-AP5) to probe dissected, unfixed third instar larvae. After subsequent fixation, DIP-α-AP5 was then detected by immunofluorescence, using antibodies against the AP epitope. In wild type animals, DIP-α-AP5 shows minimal binding to tissue (*Figure 5F*). However, upon ectopic postsynaptic expression of Dpr10-V5 in muscles, we observed strong labeling of the postsynaptic membrane surrounding boutons (*Figure 5G*), suggesting that DIP-α is able to directly bind to Dpr10. This observation supports a model whereby endogenous muscle Dpr10 interacts transsynaptically with DIP-α in MNISN-1s terminals.

## *dpr10* is required for MNISN-1s innervation of m4

Since Dpr10 interacts with DIP-α, both in vitro and in tissue, and is able to localize to the postsynaptic membrane, we decided to investigate the requirement of *dpr10* for MNISN-1s innervation of m4. In order to distinguish 1s and 1b terminals, we stained for a subsynaptic reticulum (SSR) marker, Discs-Large (Dlg), and identified 1s boutons by their less extensive SSR relative to 1b boutons (*Guan et al., 1996*) (*Figure 6A*). We examined a CRISPR-generated null deletion mutant of *dpr10*, *dpr10^{14-5}*. In this mutant, we observed loss of almost 100% of 1s boutons on m4 (*Figure 6B and D*), similar to the *DIP-α* null phenotype (*Figure 2B*). Dorsal MNISN-1s innervation (m1, 2, 9, and 10) was largely unaltered (*Figure 6—figure supplement 1A*). 1b boutons were still present on m4 (*Figure 6B*), indicating that Dpr10 is specifically required for MNISN-1s innervation.

Since *dpr6* is expressed in 1b and 1s motor neurons, and *DIP-α* is in 1s motor neurons, whose axons fasciculate with 1b axons, axon-axon contact through Dpr6–DIP-α interactions could potentially affect MNISN-1s targeting to m4. However, as shown above, MNISN-1s is able to properly defasciculate without *DIP-α* (*Figure 3C,D*). Also, *dpr6-T2A-GAL4/Df* mutants have no m4-1s phenotype, and do not produce any other detectable alterations in MNISN-1s (*Figure 6C and D*). Thus, the role of DIP-α in formation of the m4-1s branch is likely to be mediated through its interactions with Dpr10.

In the neuromuscular system, we found that *dpr10* is expressed in neurons (*Carrillo et al., 2015*) and in muscles (*Figure 5D*). To determine which cells require Dpr10 function for normal formation of the m4-1s branch, we used cell type specific RNAi knockdown of *dpr10*. Presynaptic expression of *dpr10*-RNAi with *DIP-α-GAL4* did not affect MNISN-1s targeting to m4 (*Figure 6F*). However, driving *dpr10*-RNAi in muscles (*Mef2-GAL4*) phenocopied the *dpr10* null mutation, producing an almost complete loss of m4-1s branches (*Figure 6F*). This result indicates that *dpr10* is required in muscles. Interestingly, pan-neuronal expression of *dpr10*-RNAi produced a partial loss of m4-1s (*Figure 6F*), suggesting that Dpr10 expression in a non-1s neuron also contributes to proper targeting.

We also attempted to rescue the m4-1s phenotype using the UAS-Dpr10-V5 construct described above. Dpr10-V5 localizes to NMJs when it is expressed in muscles. However, we observed that expression of Dpr10-V5 from pan-neuronal (*Elav-GAL4*) or muscle (*Mef2-GAL4*) drivers reduced the frequency of m4-1s branches, even in a wild-type background (*Figure 6E*), similar to *DIP-α* GOF. Pan-neuronal expression of Dpr10 almost eliminated m4-1s branches. We asked whether this was likely to be a *cis* or *trans* effect by expressing Dpr10 only in 1s neurons. We observed that expression of Dpr10-V5 in MNISN-1s using *DIP-α-GAL4* also produced loss of M4-1s branches (*Figure 6E*), suggesting that a GOF *cis* DIP-α–Dpr10 interaction can interfere with the DIP-α–Dpr10 *trans* interactions that are important for branch formation.

## *dpr10* is dynamically expressed in muscle subsets during embryonic development

In third instar larvae, *dpr10* is expressed in almost all abdominal muscles, but loss of branches in *DIP-α* or *dpr10* mutants is only observed for m4 and adjacent muscles. This led us to examine if *dpr10* expression in muscles was temporally and spatially regulated during the period of embryonic development in which the neuromuscular circuit is established. We used the *dpr10-GAL4>EGFP* reporter and focused on three time points: late 14/early 15, late 15/early 16, and late 16/early 17.

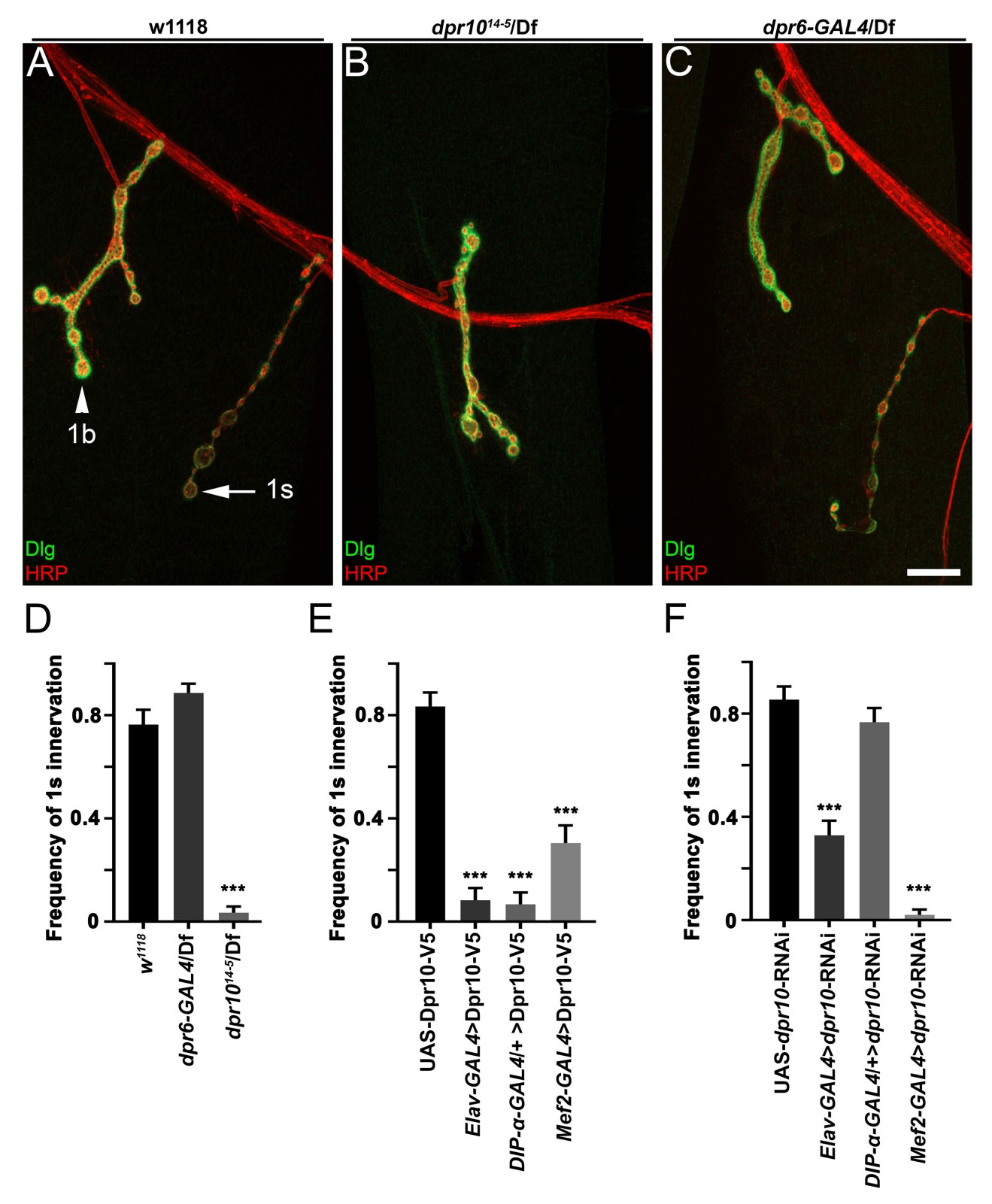

**Figure 6.** Dpr10 is required for MNISN-1s innervation of muscle 4. (A–C) Larval m4 NMJs labeled with anti-DLG (green) and anti-HRP (red) from (A) control *w1118* animals, (B) d*pr10*<sup>14-5</sup>/*Df* null mutants and (C) *dpr6-GAL4/Df*. The difference in anti-Dlg staining intensity allows us to differentiate 1b and 1s NMJs. 1b and 1s neurons are marked with arrowhead and arrow, respectively. (D–F) Frequency of 1s innervation of m4 from (D) d*pr6* and d*pr10* mutants and (E) overexpression of UAS-Dpr10-V5 and (F) UAS-*dpr10*-RNAi. n (animals/hemisegments) = (D) (8/62), (12/70), (11/60) and (E) (8/48), (6/36),

*Figure 6 continued on next page*

*Figure 6 continued*

(5/30), (8/47), (F) (8/48), (8/48), (10/60), (8/48) (respectively). Calibration bar is 16 µm in A-C. Error bars represent SEM. ***p<0.0001. See also ***Figure 6—figure supplement 1***.

DOI: https://doi.org/10.7554/eLife.42690.015

The following source data and figure supplement are available for figure 6:

**Source data 1.** Data for graphs in ***Figure 6*** and ***Figure 6—figure supplement 1***.
DOI: https://doi.org/10.7554/eLife.42690.017
**Figure supplement 1.** MNISN-1s innervation of dorsal muscles and branching on m2.
DOI: https://doi.org/10.7554/eLife.42690.016

Using confocal images of embryos stained for the axonal marker Fasciclin II (Fas2; used to more accurately determine developmental stage) and GFP, we qualitatively scored samples, where muscle GFP signal above background received a score of 1, and anything at background levels received a score of 0. ***Figure 7G*** thus represents the probability that *dpr10-GAL4>EGFP* is expressed in each muscle examined.

The first muscles that reproducibly show *dpr10* expression are m5 and m20, which flank m4 (***Figure 7A,B and G***). In late st15/early st16 embryos, *dpr10* expression becomes more consistent in m5 and m20 and expands to m2 in the dorsal muscle field and several muscles in the ventral muscle field (***Figure 7C,D and G***). By late st16/early st17, most muscles, including m4, express *dpr10* (***Figure 7E–G***). The first MNISN-1s filopodia in the vicinity of m4 are observed in late st16/early st17 embryos, and the first discernable 1s boutons on m4 are found in early first instar larvae (***Figure 3***). Thus, *dpr10* expression in m4 coincides with MNISN-1s innervation. The earlier expression of *dpr10* in muscles adjacent to m4 (m5 and m20) might mean that these muscles provide cues for formation or stabilization of an MNISN-1s axon branch in this area. Innervation of m20 is also lost in *DIP-α* mutants. m5, however, lacks 1s innervation even in wild-type. Interestingly, m18 and m11 do not express *dpr10* even in st17 embryos (***Figure 7G***). These muscles are among those that are not innervated by either of the *DIP-α* positive motor neurons (***Figure 4C***). Some of these muscles do receive 1s innervation from the third 1s motor neuron, which does not express *DIP-α*.

## Discussion

In this paper, we show that interactions between DIP-α and its in vitro binding partner, Dpr10, are essential for innervation of a specific subset of larval muscle fibers by branches of the MNISN-1s motor axon. *DIP-α* is expressed by only two motor neurons, and the protein localizes to the NMJs of those neurons (***Figure 1***, ***Figure 1—figure supplement 1***). MNISN-1s innervates most of the muscles in the dorsal muscle field, but only the proximal (most ventral) branches of its axon are affected in *DIP-α* mutants. The branch innervating m4, m4-1s, is absent in 100% of hemisegments in mutants. DIP-α is required in the MNISN-1s neuron to direct innervation of m4 (***Figure 2***). Examination of the MNISN-1s axon during embryonic development shows that its filopodia explore the surface of m4 and surrounding muscles in a normal manner in *DIP-α* mutants, but a stable m4 NMJ never forms (***Figure 3***). Innervation of muscles near m4 is also reduced in *DIP-α* mutants, while innervation of more dorsal muscles is increased (***Figure 4***). One of DIP-α's two binding partners, Dpr10, is expressed at high levels in muscles and can localize to the postsynaptic side of NMJs (***Figure 5***), and the m4-1s branch is also absent in *dpr10* mutants. RNAi knockdown experiments show that Dpr10 is required in muscles (***Figure 6***). By examination of the temporal and spatial expression patterns of *dpr10* in embryos, we found that its earliest expression is on muscles flanking m4, some of which also lack 1s NMJs in *DIP-α* mutants (***Figure 7***). This suggests that recognition of Dpr10 on these muscles by DIP-α on the MNISN-1s growth cone is a cue for branch formation or stabilization.

### A model for control of muscle innervation by interactions between DIP-α and Dpr10

A number of mutant screens for alterations in the morphologies and patterning of NMJs in the larval neuromuscular system have been performed (***Aberle et al., 2002***; ***Kraut et al., 2001***; ***Valakh et al., 2012***). LOF mutations in a few genes, including those encoding the cell-surface IgSF domain protein Sidestep and its binding partner, Beaten Path, cause motor axons to fail to arborize normally onto

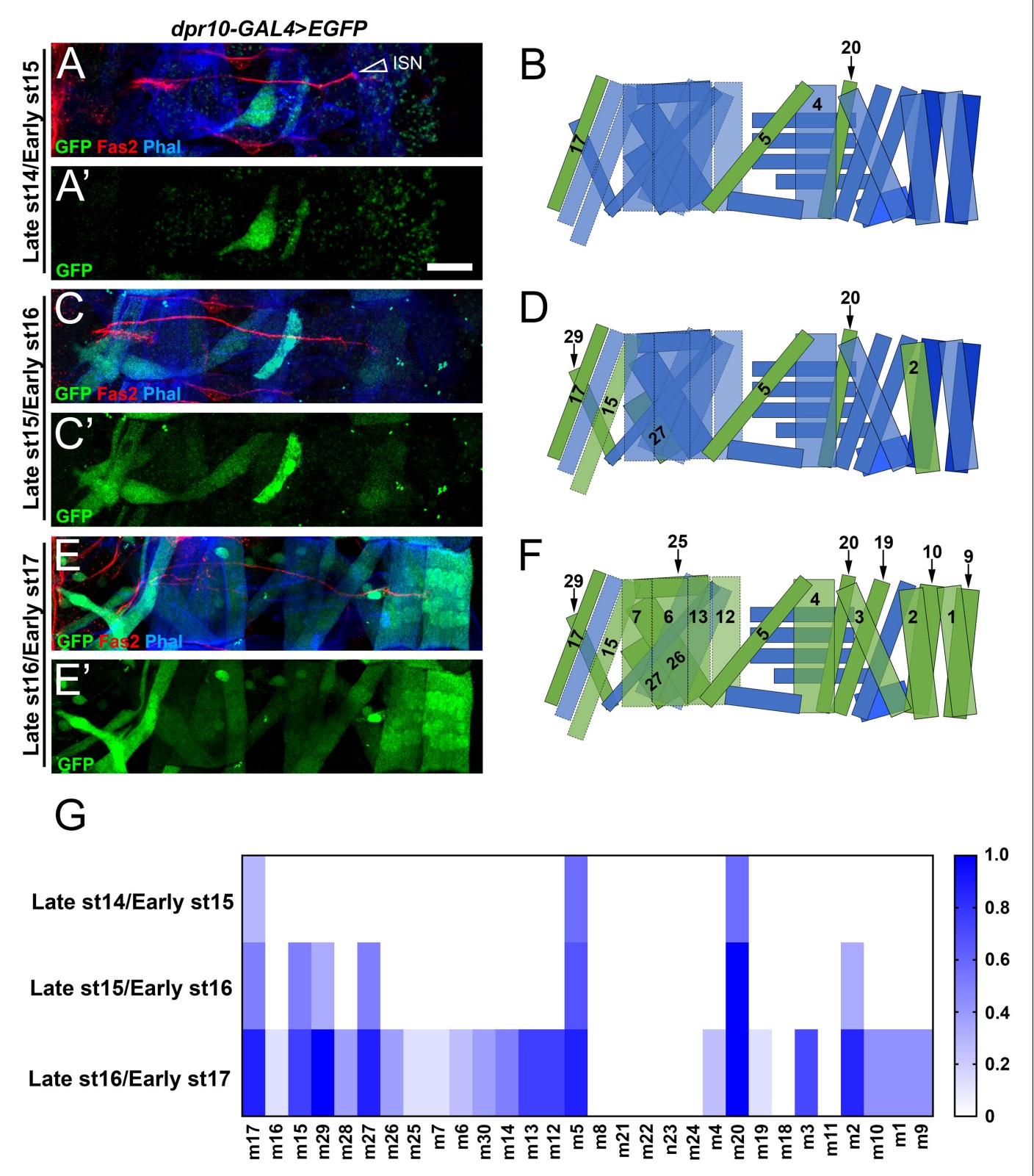

**Figure 7.** Temporal and spatial expression of *dpr10* in embryonic muscles. (**A,C,E**) Embryonic *dpr10-GAL4>EGFP* abdominal body wall hemisegments labeled with anti-GFP (green), anti-Fas2 (red) and phalloidin (blue) from (**A**) late st14/early st15, (**C**) late st15/early st16, and (**E**) late st16/early st17. Open arrowhead denotes Intersegmental Nerve (ISN), which carries dorsal neurons. (**B,D,F**) Cartoon representation of an embryonic hemisegment. Green muscles are those that express *dpr10* at the respective developmental stage while blue muscles do not express *dpr10*. (**G**) Heat map quantification of

*Figure 7 continued on next page*

*Figure 7 continued*

the frequency of *dpr10-GAL4>EGFP* expression in each muscle at each developmental stage, where 1 represents 100% probability of expression. n (embryos) = 7, 6, 8 (respectively). Calibration bar is 19 µm.

DOI: https://doi.org/10.7554/eLife.42690.018

The following source data is available for figure 7:

**Source data 1.** Data for graphs in *Figure 7*.

DOI: https://doi.org/10.7554/eLife.42690.019

any muscle fibers, resulting in large-scale alterations in innervation patterns (*Kinold et al., 2018*; *Pipes et al., 2001*; *Siebert et al., 2009*; *Sink et al., 2001*). However, there are no prior reports (to our knowledge) of LOF mutations in single genes that cause high-penetrance changes in targeting of single larval motor axons to individual, or groups of, muscle fibers.

The failure to find genes required for innervation of specific muscles in LOF screens has suggested that individual muscles may be labeled by multiple targeting cues, and that neurons express receptors for more than one of these cues. Loss of any one neuronal receptor or muscle targeting cue does not cause strong phenotypes because they have partially redundant functions. The remaining receptors and cues may substitute for the missing proteins in mutants and allow normal muscle targeting to occur.

It has been difficult to identify neuronal receptors whose expression is specific to particular subsets of motor axons. Neuronal CSPs that have been previously studied in the context of motor axon guidance and arborization onto muscles (*e.g.,* Receptor Tyrosine Phosphatases (RPTPs), Beaten Paths, Fasciclin II, Netrin receptors, Semaphorin receptors) are usually expressed by most or all motor neurons. Not surprisingly, then, mutations in genes encoding these proteins usually produce guidance or arborization phenotypes that affect many motor axons and muscles. By contrast, DIP-α is expressed in only two of the 35 motor neurons that innervate muscles in each larval abdominal hemisegment. These are the 1s motor neurons MNISN-1s (RP2) and MNISNb/d-1s. This finding suggested that any phenotypes caused by loss of DIP-α might be specific to the axons of these two motor neurons.

Like other motor axons, the two DIP-α-expressing axons probably express binding partners for many muscle cell-surface proteins. Neuronal and muscle binding partners could act as signaling receptors, ligands for receptors, or adhesion molecules. The 1s motor axons that express DIP-α have multiple branches, and each axon innervates most of the muscles within a muscle field. MNISN-1s innervates muscles in the dorsal field (*Hoang and Chiba, 2001*; *Landgraf et al., 1997*; *Nose, 2012*). One might expect that targeting phenotypes would be observed in *DIP-α* LOF mutants only if binding of DIP-α to one of its Dpr binding partners was essential for recognition of specific muscle fibers by individual branches of a 1s motor axon. In fact, we observed that the loss of DIP-α causes a high-penetrance loss of branches onto a particular group of dorsal muscle fibers innervated by MNISN-1s. These are internal m4 and m3, and external m19 and m20, which are underneath m4 and m3 in embryonic and larval fillet preparations. However, the branches of the MNISN-1s axon onto internal dorsal m2 and external m9 and m10, which lie underneath m2, are still present in *DIP-α* mutants.

MNISN-1s branches are also lost with high penetrance in *dpr10* null mutants, indicating that Dpr10 is the DIP-α binding partner relevant to innervation of these muscles. In larvae, *dpr10* is expressed at high levels in most muscle fibers. Knockdown of *dpr10* by RNAi in all muscles affects formation of the same MNISN-1s branches that are eliminated in *dpr10* mutants. Therefore, binding of neuronal DIP-α to muscle Dpr10 is likely to underlie recognition of specific muscles as targets for proximal MNISN-1s axon branches. In embryos, *dpr10* expression is initiated in m20 and m5, which flank m4. NMJs on m20 are also absent in *DIP-α* mutants; m5 does not receive 1s innervation. *dpr10* expression begins in m4 around the time at which we observe exploration of this muscle by filopodia emerging from the MNISN-1s axon (*Figure 7*).

The gene encoding DIP-α's other binding partner, Dpr6, is expressed by most motor neurons, but is not detectably expressed by muscle fibers. Although *dpr6* is expressed by MNISN-1s in embryos, *dpr6* mutants do not have m4-1s phenotypes. This suggests that Dpr6 does not play a direct role in the targeting of MNISN-1s to m4. DIP-α can also bind homophilically, but with reduced affinity relative to its heterophilic binding affinities for Dpr10 and Dpr6 (*Cheng et al., 2019*;

*Cosmanescu et al., 2018*). However, during normal development DIP-α would not have the opportunity to mediate homophilic interactions between motor axons, since it is expressed only on the MNISN-1s axon and not on the other motor axons with which it fasciculates during outgrowth.

Correct innervation of m4 and the other muscles in its immediate vicinity may require a balance between the expression levels of DIP-α's binding partner Dpr10 on muscles relative to axons. As described above, knocking down Dpr10 in muscles eliminates innervation of m4 (*Figure 6*), suggesting that transsynaptic interactions between neuronal DIP-α and muscle Dpr10 are essential for recognition of this muscle by an interstitial MNISN-1s growth cone. There is also a 50% reduction in m4 innervation when Dpr10 is knocked down in all neurons, while knockdown in MNISN-1s produces no innervation defects. This suggests that interactions between DIP-α on MNISN-1s axons and Dpr10 on other axons with which it fasciculates also contribute to correct m4 innervation.

Driving high-level expression of DIP-α or Dpr10 in all neurons abolishes m4 innervation by MNISN-1s. Normally DIP-α is not expressed in other axons in the ISN fascicle, so upon DIP-α expression ectopic axon-axon interactions mediated by homo- and heterophilic binding may alter MNISN-1s connectivity. Interestingly, these GOF phenotypes are also seen when Dpr10, but not DIP-α, is increased in MNISN-1s only. High-level expression of DIP-α or Dpr10 on muscles also eliminates (DIP-α) or reduces (Dpr10) innervation of m4. Some of these phenotypes may be due to *cis* Dpr10–DIP-α interactions on the same membrane, which could reduce the amount of DIP-α or Dpr10 that is available to interact with its partner in trans. Excessive adhesion between the MNISN-1s axon and the other axons in its bundle (in the case of Dpr10 overexpression in all neurons), or between the MNISN-1s axon and the muscles it traverses during its outgrowth (in the case of Dpr10 overexpression in muscles) may also affect the ability of a branch to separate from the axon and form an NMJ. Overexpression of Dpr10 in muscles may similarly cause excessive MNISN-1s adhesion to distal muscles, and this model is supported by the exuberant number of branches on m2 (*Figure 6—figure supplement 1B*).

Knockdown or overexpression of DIP-α or Dpr10 in neurons or in muscles does not reduce the frequency of innervation of the most dorsal muscles by MNISN-1s, indicating that these muscles are recognized as targets *via* other cues. Interestingly, however, m1, which is adjacent to m2 and rarely innervated by MNISN-1s, gains innervation in *DIP-α* mutants, and the 1s NMJ on m2 becomes larger. These results suggest that MNISN-1s is normally specified to make a certain number of synaptic boutons, and that loss of boutons on proximal muscles m4, m3, m19, and m20 results in an increased number of boutons on distal muscles.

Using these results, we have constructed a model that can explain how interactions between DIP-α and Dpr10 specify targeting of MNISN-1s axon branches to m4 and the other muscles in its vicinity. DIP-α begins to be expressed in MNISN-1s (RP2) in st14 embryos, during the period of motor axon guidance. The MNISN-1s axon reaches its terminus in the vicinity of m1/m2 at st16, before it forms interstitial branches onto m20, where Dpr10 is already expressed. After the m20-1s branch forms, Dpr10 appears on m4, and binding of DIP-α on MNISN-1s to Dpr10 on m4 and surrounding muscles results in the formation of stable branches that differentiate into NMJs. During this process, DIP-α on MNISN-1s might switch from interacting with Dpr10 on fasciculated axons within the ISN bundle to binding to Dpr10 on muscles.

## Functions of the Dpr-DIP network in formation of synaptic circuits

The Dpr-ome binding network was defined by a global in vitro 'interactome' screen for binding interactions among all *Drosophila* cell-surface and secreted proteins containing three common extracellular domain types: IgSF, Fibronectin Type III, and LRR. There are 21 Dpr proteins, each containing two IgSF domains, 11 DIP proteins, each containing three IgSF domains, and an LRR protein called cDIP that binds to many Dprs and DIPs (*Carrillo et al., 2015*; *Cosmanescu et al., 2018*; *Özkan et al., 2013*).

Analysis of expression of individual *dpr* and *DIP* genes revealed remarkable and unprecedented patterns in the larval ventral nerve cord and pupal brain. Each *dpr* and *DIP* is expressed by a small and unique subset of interneurons. In the pupal optic lobe, neurons expressing a particular Dpr are often presynaptic to neurons expressing a DIP to which that Dpr binds in vitro (*Carrillo et al., 2015*; *Tan et al., 2015*; *Xu et al., 2018*). These findings suggested that Dpr-DIP interactions might be important for formation of synaptic circuits during brain and ventral nerve cord development.

In our earlier work, we examined the expression and function of DIP-γ and its binding partner Dpr11. Dpr11 is selectively expressed in 'yellow' R7 photoreceptors, which make the Rh4 rhodopsin, and DIP-γ is expressed in a subset of Dm8 amacrine neurons in the optic lobe medulla. Dm8s receive more synapses from R7 than any other neuron. DIP-γ is required for survival of the Dm8 neurons that express it (*Carrillo et al., 2015*). The fact that loss of DIP-γ causes loss of brain neurons that express these proteins suggests that DIP–Dpr interactions can transmit trophic signals. This does not appear to be the case for either DIP in the larval or adult neuromuscular system, however, since there are no missing motor neurons in *DIP-γ* or *DIP-α* mutants.

The expression patterns of DIP-γ and DIP-α suggest that they may be involved in similar processes during optic lobe development. In addition to yellow R7s, Dpr11 is expressed in a subset of motion-sensitive T4 and T5 neurons, which synapse onto large cells called Lobula Plate Tangential Cells (LPTCs). Dpr11-expressing T4 and T5 cells project to the layers 1 and 2 of the lobula plate, and DIP-γ is expressed in a small number of LPTCs that arborize in those layers (*Carrillo et al., 2015*). In the optic lobe lamina, L3 and L5 neurons express Dprs 6 and 10, while L2 expresses only Dpr6. These L cells are synaptically connected to Dm4, Dm12, and Dm1 cells in the medulla, which express DIP-α (*Tan et al., 2015*). Loss of DIP-α or of both Dprs 6 and 10 causes death of some Dm4 neurons and affects synaptic targeting of Dm12 neurons (*Xu et al., 2018*).

In the larval neuromuscular system, however, the functions of DIP-γ appear to be very different from those of DIP-α. DIP-γ and Dpr11 are both expressed by most or all motor neurons. In *DIP-γ* and *dpr11* LOF mutants, there are no alterations in muscle targeting, but NMJs have phenotypes characterized by the presence of small clustered boutons called satellites. Retrograde BMP signaling is upregulated in these mutants (*Carrillo et al., 2015*). By contrast, DIP-α is expressed by only two motor neurons, and its interactions with Dpr10 expressed on muscles control formation and/or targeting of a specific set of interstitial axon branches.

The functions of DIP-α and Dpr10 appear to be conserved between the larval neuromuscular system and the adult leg neuromuscular system. The accompanying paper from Richard Mann's group (*Venkatasubramanian et al., 2019*) shows that DIP-α is expressed in a subset of motor neurons that innervate specific leg muscles, while Dpr10 is expressed in muscles. In *DIP-α* and *dpr10* mutants, the axonal branches onto the muscles targeted by the DIP-α-expressing axons are absent. In summary, Dpr10 appears to be one of the long-sought targeting cues that direct recognition of specific muscle fibers as targets, while DIP-α is the corresponding receptor on the motor neurons that innervate these muscles.

# Materials and methods

### Key resources table

| Reagent type (species) or resource | Designation | Source or reference | Identifiers | Additional information |
|---|---|---|---|---|
| Genetic reagent (*D. melanogaster*) | *w^1118* | | Bloomington Drosophila Stock center (BDSC) | |
| Genetic reagent (*D. melanogaster*) | *Mef2-GAL4* | PMID: 9671578 | | Gift of Hugo Bellen |
| Genetic reagent (*D. melanogaster*) | *DIP-α-T2A-GAL4* | PMID: 21985007 | | Gift of Hugo Bellen |
| Genetic reagent (*D. melanogaster*) | *DIP-α-EGFP-DIP-α* | PMID: 26687361 | | Gift of Hugo Bellen |
| Genetic reagent (*D. melanogaster*) | *DIP-α^1-178* | PMID: 30467079 | | Gift of Lawrence Zipursky |
| Genetic reagent (*D. melanogaster*) | *dpr10^14-5* | PMID: 30467079 | | Gift of Lawrence Zipursky |
| Genetic reagent (*D. melanogaster*) | UAS-DIP-α-Myc | PMID: 30467079 | | Gift of Lawrence Zipursky |
| Genetic reagent (*D. melanogaster*) | UAS-Dpr10-V5 | PMID: 30467079 | | Gift of Lawrence Zipursky |

*Continued on next page*

*Continued*

| Reagent type (species) or resource | Designation | Source or reference | Identifiers | Additional information |
|---|---|---|---|---|
| Genetic reagent (*D. melanogaster*) | *dpr10-T2A-GAL4* | PMID: 21985007 | | Gift of Hugo Bellen |
| Genetic reagent (*D. melanogaster*) | *dpr6-T2A-GAL4* | PMID: 21985007 | | Gift of Hugo Bellen |
| Genetic reagent (*D. melanogaster*) | UAS-2XEGFP | PMID: 12324968 | BDSC #6874 | |
| Genetic reagent (*D. melanogaster*) | *Elav-GAL4* | PMID: 22319582 | BDSC #8765 | |
| Genetic reagent (*D. melanogaster*) | 24B-*GAL4* | PMID: 8223268 | BDSC #1767 | |
| Genetic reagent (*D. melanogaster*) | *MHC*-GeneSwitch | PMID: 11675495 | | Gift of Haig Keshishian |
| Genetic reagent (*D. melanogaster*) | $Eve^{RN2}$-*GAL4* | PMID: 14624243 | BDSC #7470 | |
| Genetic reagent (*D. melanogaster*) | UAS-*dpr10*-RNAi | PMID: 17625558 | Vienna Drosophila Resource Center #103511 | |
| Genetic reagent (*D. melanogaster*) | UAS-*DIP-α*-RNAi | PMID: 26320097 | BDSC #38965 | |
| Genetic reagent (*D. melanogaster*) | Df(3L)BSC673 | | BDSC #26525 | Deficiency covering dpr6 and dpr10 |
| Antibody | Goat anti-HRP-TRITC | Jackson Immunological Research | #123-025-021 | 1:50 |
| Antibody | Goat anti-HRP-Alexa405 | Jackson Immunological Research | #123-475-021 | 1:50 |
| Antibody | Mouse anti-Dlg | Developmental Studies Hybridoma Bank | #4F3 | 1:100 |
| Antibody | Mouse anti-Fas2 | Developmental Studies Hybridoma Bank | #1D4 | 1:100 |
| Antibody | Mouse anti-DIP-α | PMID: 30467079 | Gift of Lawrence Zipursky | 1:20 |
| Antibody | Mouse anti-V5 | ThermoFisher | #R960-25 | 1:400 |
| Antibody | Rabbit anti-GFP | ThermoFisher | #A11122 | 1:1000 |
| Antibody | Rabbit anti-Dlg | PMID: 9354326 | Gift of Vivian Budnik | 1:40,000 |
| Antibody | Rabbit anti-Myc | Cell Signaling Technology | #71D10 | 1:200 |
| Antibody | Goat anti-Mouse-Alexa488 | ThermoFisher | #A11029 | 1:500 |
| Antibody | Goat anti-Mouse-Alexa568 | ThermoFisher | #A11031 | 1:500 |
| Antibody | Goat anti-Rabbit-Alexa488 | ThermoFisher | #A11008 | 1:500 |
| Antibody | Goat anti-Rabbit-Alexa568 | ThermoFisher | #A11036 | 1:500 |
| Chemical compound, drug | Phalloidin-Alexa647 | ThermoFisher | #A22287 | 1:100 |
| Antibody | Rabbit anti-alkaline phosphatase | Abcam | #ab16695 | 1:100 |

## Transgenic and T2A-Gal4 lines

For construction of UAS-DIP-α-Myc, the Myc sequence (gaacaaaaacttatttctgaagaagatctg) was inserted two amino acids before the end of DIP-α protein sequence, following a GGS linker sequence, and cloned into JFRC28 vector. The final construct was injected into BDSC stock #9744, and inserted into chromosome 3R via PhiC31 mediated integration (Bestgene, Inc.). Plasmid and primer design were carried out using the software Snapgene.

T2A-*GAL4* lines were generated as described in (*Diao et al., 2015*). Briefly, flies carrying the MiMIC insertion were crossed with the flies bearing the triplet donor (T2A-*GAL4* for all three phases of DNA). The $F_1$ males from this progeny carrying both components were then crossed to females carrying germline transgenic sources of Cre and ΦC31. The $F_2$ males with all four components were crossed to a UAS-2XEGFP reporter line and progeny larvae were screened for T2A-*GAL4* transformants. The GFP+ larvae were confirmed by PCR.

## Dissection and immunocytochemistry

Embryonic dissections were performed as in (*Lee et al., 2009*). Egg laying chambers were setup with adult flies and grape juice plates (3% agar, 1.3% sucrose, 25% grape juice in water) and left in the dark at room temperature to lay eggs for 2 hr. Embryos were incubated overnight at 18°C, and then raised to 29°C for 2 hr to maximize GFP expression. Under a Zeiss V20 stereoscope with fluorescence, embryos were transferred to a microscope slide with double sided tape and staged using the autofluorescence and shape of the gut (*Bownes, 1975*; *Hartenstein et al., 1987*). Embryos were dechorionated on the double sided tape with a sharpened metal probe and transferred to a small agar slab. A dissection chamber was then prepared on Superfrost Plus slides (ThermoFisher #22-037-246) by outlining a rectangle with a PAP pen (Research Products International, #195506) to create a dissecting area. Embryos were transferred to a small piece of double sided tape placed into the dissection area, and then covered with phosphate buffered saline (PBS) (0.01M Sodium Phosphate, 150 mM Sodium Chloride). Using a 0.1 mm electrolytically sharpened tungsten wire held in a pin vice, the embryos were carefully opened along their dorsal surface, and removed from the vitellin membrane. Dissected embryos were then trasferred to the adherent charged slide. When all embryos were dissected, the well was washed once with PBS, and then fixed for 1 hr at room temperature using 4% paraformaldehyde (20% paraformaldehyde solution (Electron Microscopy Sciences) diluted into PBS). After fixation, samples were washed three times, 15 min each, in 0.05% PBST (PBS with 0.05% TritonX100), and then blocked for 1 hr in 5% normal goat serum (5% Goat serum in 0.05% PBST). Slides were incubated in primary antibody solutions overnight at 4°C in a humidified chamber. Samples were then washed three times, 15 min each, with PBST, and incubated for two hours at room temperature with secondary antibodies. After incubation, samples were washed three times, 15 min each, with PBST and mounted in vectashield (Vector Laboratories).

Larval dissections were performed as in (*Menon et al., 2015*). Wandering third instar larvae were dissected in PBS on Sylgard dishes, and pinned down using 0.1 mm Insect Pins (FST #26002–10). Samples were then fixed for 30 min using 4% paraformaldehyde (see above). After fixation samples were washed three times, 15 min each, in PBST. Samples were then blocked for 1 hr in 5% normal goat serum (5% Goat serum in 0.05% PBST), and then incubated in primary antibodies overnight at 4°C. On the second day, the larval preps were treated as embryo samples above. All larval washes and antibody incubations were performed with mild agitation on a nutator.

For in vivo staining of larvae (*Figure 5F,G*), we modified a previously described embryo protocol in (*Bali et al., 2016*; *Fox and Zinn, 2005*; *Lee et al., 2013*). Supernatant from S2 cell culture containing the DIP-α-$AP_5$ fusion protein was concentrated 5-fold (Amicon Ultra-4 Filter Unit, 100 kDa cutoff). After larvae were dissected in sylgard dishes, the area surrounding each larva was patted dry with a Kimwipe. The DIP-α-$AP_5$ (75 µl) was applied to each prep and incubated for 1.5 hr at room temperature in a humidified chamber. Following incubation, the standard larval immunocytochemistry protocol (above) was followed.

## Microscopy and analysis

Third instar samples that were quantified for 1s bouton presence or absence were examined using a Zeiss AxioImager M2 and a 40X plan-neofluar 1.3NA objective. Boutons were examined using HRP and scored as 1s or 1b based on Dlg signal, as 1s boutons have a smaller and dimmer Dlg signal

than 1b boutons. For *DIP-α-GAL4*>EGFP studies, 1s boutons were scored based on the presence of GFP although Dlg staining was also present. To count embryonic MNISN-1s terminal swellings, we grouped dorsal muscles into the distal (m1, 2, 9, 10) and proximal (m3, 4, 19, 20) fields and scored bulbous endings that were at least twice the thickness of the neurite shaft. Primary and secondary branches on m2 were scored and the total reported.

All imaging was done on a Zeiss LSM800 confocal microscope with 20X plan-apo 0.8NA objective, 40X plan-neofluar 1.3NA objective, or 63X plan-apo 1.4NA objective. All samples were imaged under similar conditions. Some embryo images were imaged using a Zeiss LSM880 with Airyscan to improve resolution of small axonal processes.

For *dpr10-GAL4*>EGFP embryonic analysis, LSM800 confocal images were acquired from embryos labelled with Fas2 (to visualize embryonic stage and nerve processes), GFP (to confirm Dpr10 expression), and Phalloidin (to visualize the muscles). Embryonic stage was determined by the number of Fas2 positive longitudinal nerves on either side of the ventral nerve cord midline (1–1.5 longitudinal nerves = st14/15, 1.5–2 = st15/16, 2.5–3 = st16/17). Each muscle was scored 1 or 0 based on the presence or absence of GFP expression, respectively, and the resulting numbers pooled into a table. The mean of each muscle subtype was then calculated, and the resulting numbers plotted in a heat map using Prism seven software (Graphpad Software).

All statistical analysis was performed using Prism seven software (Graphpad), for multiple comparisons one-way ANOVA analysis was performed with Dunnett's test. For each data point at least three samples per genotype were dissected and at least two biological replicates were examined.

## Acknowledgements

We thank Larry Zipursky for sharing unpublished *dpr10* and *DIP-α* CRISPR and UAS transgenic lines and Hugo Bellen for *dpr6-T2A-GAL4*, *dpr10-T2A-GAL4*, and *DIP-α-T2A-GAL4*. Transgenic UAS-*dpr10*-RNAi stock was obtained from the Vienna Drosophila Resource Center (VDRC, www.vdrc.at). Stocks obtained from the Bloomington Drosophila Stock Center (NIH P40OD018537) were used in this study. The monoclonal antibodies, 4F3, 1D4 developed by Goodman, C. (University of California, Berkeley) were obtained from the Developmental Studies Hybridoma Bank, created by the NICHD of the NIH and maintained at The University of Iowa, Department of Biology, Iowa City, IA 52242. We thank Ellie Heckscher, Maria Plutarco, Larry Zipursky, Richard Mann, and Lalanti Venkatasubramanian for helpful discussions and comments. This work was supported by National Institutes of Health Grants K01 NS102342 (to RAC), R01 NS096509 (to KZ), R01 NS097161 (subcontract to KZ from Engin Özkan), and T32 GM007183 (to ML-R), and funding from the University of Chicago BSD Office of Diversity and Inclusion (to RAC) and the Grossman Institute for Neuroscience, Quantitative Biology and Human Behavior (to RAC).

## Additional information

### Funding

| Funder | Grant reference number | Author |
| --- | --- | --- |
| National Institute of General Medical Sciences | T32 GM007183 | Meike Lobb-Rabe |
| National Institute of Neurological Disorders and Stroke | R01 096509 | Kai Zinn |
| National Institute of Neurological Disorders and Stroke | K01 NS102342 | Robert A Carrillo |
| University of Chicago Grossman Institute for Neuroscience, Quantitative Biology and Human Behavior | | Robert A Carrillo |
| University of Chicago BSD Faculty Diversity Career Advancement Grant | | Robert A Carrillo |

The funders had no role in study design, data collection and interpretation, or the decision to submit the work for publication.

## Author contributions

James Ashley, Conceptualization, Formal analysis, Validation, Investigation, Visualization, Methodology, Writing—original draft, Writing—review and editing; Violet Sorrentino, Formal analysis, Investigation, Writing—review and editing; Meike Lobb-Rabe, Data curation, Formal analysis; Sonal Nagarkar-Jaiswal, Resources, Writing—review and editing; Liming Tan, Shuwa Xu, Qi Xiao, Resources; Kai Zinn, Conceptualization, Supervision, Funding acquisition, Writing—original draft, Writing—review and editing; Robert A Carrillo, Conceptualization, Resources, Formal analysis, Supervision, Funding acquisition, Investigation, Methodology, Writing—original draft, Project administration, Writing—review and editing

## Author ORCIDs

James Ashley (iD) http://orcid.org/0000-0002-6693-1014
Kai Zinn (iD) http://orcid.org/0000-0002-6706-5605
Robert A Carrillo (iD) http://orcid.org/0000-0002-2067-9861

## Decision letter and Author response

Decision letter https://doi.org/10.7554/eLife.42690.022
Author response https://doi.org/10.7554/eLife.42690.023

# Additional files

## Supplementary files

• Transparent reporting form
DOI: https://doi.org/10.7554/eLife.42690.020

## Data availability

All data generated or analysed during this study are included in the manuscript and supporting files.

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
