## [Decision Letter]

Thank you for submitting your article "Transsynaptic interactions between IgSF proteins DIP-α and Dpr10 are required for motor neuron targeting specificity" for consideration by *eLife*. Your article has been reviewed by K VijayRaghavan as the Senior Editor, a Reviewing Editor, and three reviewers. The following individual involved in the review of your submission has agreed to reveal their identity: Durafshan Sakeena Syed (Reviewer #1).

The reviewers have discussed the reviews with one another and the Reviewing Editor has drafted this decision to help you prepare a revised submission.

Summary:

Carillo, Zinn and others have previously characterized two groups of interacting immunoglobulin superfamily adhesion proteins, DIPs and dprs, and implicated them in synaptic development in the *Drosophila* neuromuscular and visual systems. Here, they provide an in-depth analysis of one pair in one system – DIP-α, expressed in a small group of larval motor neurons, and dpr10, expressed in a set of larval muscles. They analyze expression of the genes and localization of the proteins, then use genetic methods to demonstrate that interactions of DIP-α with dpr10 are required for specific connectivity of the DIP-α-expressing neurons with some of the dpr10-expressing muscles. The idea that pre- and postsynaptic cells express interacting adhesion molecules that play critical roles in the formation of a specific synaptic connection is a widely accepted dogma. Yet decisive evidence for this model is sparse. The importance of this paper is that it provides such compelling evidence. The paper is through, including detailed analysis of expression, documentation of global effects on connectivity in the relevant muscle group (some muscles gain inputs at the expense of those that lose it in mutants), proof that DIP-α acts pre- and dpr10 postsynaptically, and consideration of the stages and steps at which the critical interactions occur. It is well-illustrated, the results are thorough, if excessively, described, and the few caveats are considered and discussed in detail. Although the conclusion is not entirely novel, this really is a textbook demonstration of an extremely important concept in developmental neurobiology. Its publication will be important for the field.

Essential revisions:

The following comments should be speedily addressed before acceptance. Comments 1-3: additional data, if available or speedily doable, will be welcome. Else, the points may simply be discussed.

1) DIP-α is expressed in only 2 out of the 35 MNs: MNISN-1s that innervates the dorsal muscles and MNSNb/d-1s that innervates the ventral muscles. How does the innervation pattern of MNSNb/d look like in DIP-α mutants?

2) Authors use DIP-α protein trap to reveal DIP-α localization in 1s boutons. Is it localized to boutons of all or specific muscle fibers? What is the expression profile of DIP-α during development? Is it localized to the growth cones or all over the axon during earlier stages?

3) Filopodia for most dorsal muscles are evident at earlier stages and later in the proximal ones (Figure 3). Are the boutons formed simultaneously in different muscle fibers or first in the dorsal-most? Does it imply that the branches are formed progressively from the distal-most region of an axon to the proximal end?

4) Authors mention that MNISN-1s innervates the dorsal muscles, the most distal of which is m1 (subsection “DIP-α does not control MNISN-1s axon guidance or defasciculation”), and at St15 MNISN-1s growth cone is visible in the m1/m2 target area (Figure 3). Later on, they show that MNISN-1s does not innervate m1 (Figure 4). If that is the case, then the growth cone visible at St15 might belong to m9 rather than m1, then it should be referred to as m2/m9 target area.

5) Authors show a differential effect on MNISN-1s innervation in DIP-α mutants. More dorsal muscles in the dorsal muscle field gain 1s branches, while more ventral muscles lose innervation. Among the dorsal-most muscles, m9 and m10 seem oblivious to DIP-α expression, while as m1 and m2 gain innervation. All these muscle fibers express Dpr10. Other dorsal-most muscles, that gain innervation are m11 and m18, do not express Dpr10. More ventral muscles (m3, 4, 19, 20) lose innervation; all these fibers express Dpr10. Although the innervation loss could be explained by the loss of adhesion, how is the gain in innervation explained? Does this phenotype also imply that the branching starts from the distal-most regions of the axon and proceeds towards the proximal end? How is the number of branches specified?

6) In the Dpr10 mutants, how is the MNISN-1s innervation in dorsal-most muscles?

7) Subsection “DIP-α is selectively expressed by two identified motor neurons”: For completeness, what are the non-MN DIP-α cells.

8) Figure 2B: Where are the 1b endings?

9) Subsection “Dpr10 is able to bind DIP-α in tissue”: Presumably lack of staining in wild-types reflects limited sensitivity of the DIP-α-AP probe but this should be made explicit, along with some estimate of the degree of overexpression needed for detection. There is also a bit of a conundrum here: It seems that considerable overexpression is required to detect the dpr10, yet it remains confined to synaptic sites even though one might expect that the sites would be saturated and excess dpr10 would be present extrasynaptically. Can the authors provide an explanation.

10) Figure 5E: Need a control (wild-type?) for the V5 staining.

11) The data is well documented, except in the case of Dpr10. In the Dpr10 case to identify its spatiotemporal expression authors use Dpr10-Gal4 reporter line only. This is surprising as this GAL4 line (like many others) may not fully reproduce endogenous gene expression. It is essential to use other tools (in situ hybridization if antibodies are not available) to show how Dpr10 is really expressed. Consequently, the validity of all author's comments related to Dpr10 expression and its correlation with that of DIP-α depends on this analysis.

---

## [Author Response]

Essential revisions:The following comments should be speedily addressed before acceptance. Comments 1-3: additional data, if available or speedily doable, will be welcome. Else, the points may simply be discussed.1) DIP-α is expressed in only 2 out of the 35 MNs: MNISN-1s that innervates the dorsal muscles and MNSNb/d-1s that innervates the ventral muscles. How does the innervation pattern of MNSNb/d look like in DIP-α mutants?

We have examined MNSNb/d-1s in *DIP-α* mutants, and we found no obvious high penetrance loss of innervation of any of its target muscles. We add a line to the text to highlight that there is no highly penetrant phenotype upon loss of *DIP-α* (subsection “DIP-α is required for MNISN-1s targeting specificity”).

2) Authors use DIP-α protein trap to reveal DIP-α localization in 1s boutons. Is it localized to boutons of all or specific muscle fibers? What is the expression profile of DIP-α during development? Is it localized to the growth cones or all over the axon during earlier stages?

We will address this comment with additional experiments. We determined that DIP-α localizes to boutons on all muscles and include an additional panel, Figure 1—figure supplement 1E (subsection “*DIP-α* is selectively expressed by two identified motor neurons”). DIP-α seems to be continuously expressed throughout embryonic and larval development, based on our analysis of *DIP-α* gene trap and protein trap lines. In the current Figure 1 and Figure 3, we show three stages of development, and DIP-α is expressed at each stage. We explicitly state in the text that DIP-α is expressed during embryonic st14 through the end of larval development (subsection “*DIP-α* is selectively expressed by two identified motor neurons” and subsection “DIP-α does not control MNISN-1s axon guidance or defasciculation”). Finally, we dissected and imaged *DIP-α* protein trap embryos and determined that DIP-α localizes to growth cones. This data is included as an additional panel, Figure 1D (subsection “*DIP-α* is selectively expressed by two identified motor neurons”).

3) Filopodia for most dorsal muscles are evident at earlier stages and later in the proximal ones (Figure 3). Are the boutons formed simultaneously in different muscle fibers or first in the dorsal-most? Does it imply that the branches are formed progressively from the distal-most region of an axon to the proximal end?

Based on our observations, it seems that MNISN-1s first contacts dorsal muscles before making contact with more proximal muscles. To address this thoroughly, a detailed time-lapse analysis is required. This level of detail is beyond the scope of this manuscript, but to provide additional data to support our observations, we focused on st16/17 embryos and scored the appearance of motor neuron terminal ‘swellings’ in the distal vs. proximal muscle field. A bar graph is added to Figure 3—figure supplement 1B (subsection “DIP-α does not control MNISN-1s axon guidance or defasciculation”).

4) Authors mention that MNISN-1s innervates the dorsal muscles, the most distal of which is m1 (subsection “DIP-α does not control MNISN-1s axon guidance or defasciculation”), and at St15 MNISN-1s growth cone is visible in the m1/m2 target area (Figure 3). Later on, they show that MNISN-1s does not innervate m1 (Figure 4). If that is the case, then the growth cone visible at St15 might belong to m9 rather than m1, then it should be referred to as m2/m9 target area.

We thank the reviewer for bringing this to our attention. However, when selecting a representative image to include in Figure 4, it was very difficult to obtain an MNISN-1s that has the perfect innervation pattern. m1 is innervated infrequently by MNISN-1s (Figure 4C) so the growth cone in Figure 3 could be m1/m2 or m2/m9. We prefer to keep the m1/m2 designation.

5) Authors show a differential effect on MNISN-1s innervation in DIP-α mutants. More dorsal muscles in the dorsal muscle field gain 1s branches, while more ventral muscles lose innervation. Among the dorsal-most muscles, m9 and m10 seem oblivious to DIP-α expression, while as m1 and m2 gain innervation. All these muscle fibers express Dpr10. Other dorsal-most muscles, that gain innervation are m11 and m18, do not express Dpr10. More ventral muscles (m3, 4, 19, 20) lose innervation; all these fibers express Dpr10. Although the innervation loss could be explained by the loss of adhesion, how is the gain in innervation explained? Does this phenotype also imply that the branching starts from the distal-most regions of the axon and proceeds towards the proximal end? How is the number of branches specified?

In the Discussion section, we propose that MNISN-1s may be programmed to form a certain number of connections (boutons and/or active zones). In the absence of *DIP-α*, appropriate connections are not formed, and this could lead to gain of innervations, or branches, at inappropriate and appropriate muscles, even if the muscles do not express Dpr10 (m11 and m18). DIP-α-Dpr10 interactions may be required to stabilize branches on specific muscles, but the underlying mechanism of branch formation is not well understood and beyond the scope of this study. However, we counted branches on m2 in *DIP-α* mutants and controls, and include an additional graph in Figure 4—figure supplement 1A (subsection “Loss of DIP-α selectively reduces and expands MNISN-1s connectivity”).

6) In the Dpr10 mutants, how is the MNISN-1s innervation in dorsal-most muscles?

We quantified innervation of m1, m2, m9, and m10 and include a graph in Figure 6—figure supplement 1B (subsection “*dpr10* is required for MNISN-1s innervation of M4”).

7) Subsection “DIP-α is selectively expressed by two identified motor neurons”: For completeness, what are the non-MN DIP-α cells.

While it would be interesting to identify what other cells express DIP-α in the VNC, it is beyond the scope of this study. In Carrillo et al., 2015, we show that a subset of these cells are cholinergic interneurons.

8) Figure 2B: Where are the 1b endings?

We should have been more diligent in labeling of this Figure. The 1B ending is present but we did not label it. We have updated this Figure with the appropriate labels.

9) Subsection “Dpr10 is able to bind DIP-α in tissue”: Presumably lack of staining in wild-types reflects limited sensitivity of the DIP-α-AP probe but this should be made explicit, along with some estimate of the degree of overexpression needed for detection. There is also a bit of a conundrum here: It seems that considerable overexpression is required to detect the dpr10, yet it remains confined to synaptic sites even though one might expect that the sites would be saturated and excess dpr10 would be present extrasynaptically. Can the authors provide an explanation.

Dpr10-V5 seems to be targeted specifically to the postsynapse based on anti-V5 immunohistochemistry (Figure 5E) and the DIP-α-AP probe (Figure 5G). We observe minimal extrasynaptic Dpr10 upon overexpression. There are several reasons why this may be the case. First, the extent of overexpression may not be sufficient to saturate the postsynaptic site. Also, staining with anti-V5 is more sensitive than the DIP-α-AP probe so any extrasynaptic Dpr10-V5 would more likely be detectable with anti-V5; however, we do not observe this (Figure 5E). Thus, if excess Dpr10 does not localize properly to the postsynapse, it may be degraded.

10) Figure 5E: Need a control (wild-type?) for the V5 staining.

This was an oversight on our part, as the anti-V5 labeling in control animals is almost nonexistent. This is now included in Figure 5—figure supplement 1B (subsection “Expression of DIP-α binding partners, Dpr6 and Dpr10, in the neuromuscular system”).

11) The data is well documented, except in the case of Dpr10. In the Dpr10 case to identify its spatiotemporal expression authors use Dpr10-Gal4 reporter line only. This is surprising as this GAL4 line (like many others) may not fully reproduce endogenous gene expression. It is essential to use other tools (in situ hybridization if antibodies are not available) to show how Dpr10 is really expressed. Consequently, the validity of all author's comments related to Dpr10 expression and its correlation with that of DIP-α depends on this analysis.

It is true that many GAL4 lines do not recapitulate endogenous gene expression. However, the *dpr10-T2A-GAL4* line that we utilize is a direct gene trap insertion within a *dpr10* coding intron derived from a MiMIC insertion (see Venken et al., 2011; Diao et al., 2015). In T2A-*GAL4* lines in coding introns, GAL4 can only be expressed when translation of the gene product (Dpr10, in this case) is initiated from its normal ATG. The T2A-*GAL4* insertion element is aberrantly spliced into the *dpr10* transcript due to the strong splice acceptor in the element. This hybrid *dpr10* transcript is then translated from the normal Dpr10 ATG to the T2A element, which then directs ribosome skipping and reinitiation, so as to generate GAL4. Therefore, GAL4 will be made under control of all endogenous *dpr10* transcriptional control elements, and will only be produced when Dpr10 translation is initiated from the normal ATG. It should therefore accurately reflect normal Dpr10 protein expression, except for three issues: (1) the reporter will not display the same subcellular localization as the protein; (2) there will be a delay caused by the time required for GAL4 to activate reporter expression and for reporter to be translated; (3) the reporter will not be subject to any post-translational control of Dpr10 expression, such as protein degradation, so it may have a different lifetime than normal Dpr10. In situ hybridization experiments could provide information about *dpr10* mRNA expression, but they would not add anything to the conclusions we can draw from the GAL4 reporter. In situs report the presence of RNA, which does not necessarily represent protein expression, so they provide less information than T2A-*GAL4* reporters. A better experiment, as the reviewer suggests, would be immunohistochemistry with a Dpr10 antibody. However, since this antibody is a polyclonal mouse antibody (see Xu et al., 2018), we were only able to obtain a very small quantity from the Zipursky lab. We have not utilized this antibody for embryo immunohistochemistry so some optimization is required to obtain reproducible data. Also, the detailed developmental studies (Figure 7) will require many samples at several embryonic stages to accurately depict Dpr10 temporal expression, and the antibody may not be sufficient. We hope that this explanation is sufficient to address the reviewers concerns.